# Application of High-Dimensional Fuzzy K-means Cluster Analysis to CALIOP/CALIPSO Version 4.1 Cloud-Aerosol Discrimination

Shan Zeng[1,2], Mark Vaughan[2], Zhaoyan Liu[2], Charles Trepte[2], Jayanta Kar[1,2], Ali Omar[2], David Winker[2], Patricia Lucker[1,2], Yongxiang Hu[2], Brian Getzewich[1,2], and Melody Avery[2]

[1] Science Systems and Applications, Inc., Hampton, 23666, USA
[2] NASA Langley Research Centre, Hampton, 23666, USA

*Correspondence to*: Shan Zeng (shan.zeng@ssaihq.com)

**Abstract.** This study applies Fuzzy K-Means (FKM) cluster analyses to a subset of the parameters reported in the CALIPSO lidar level 2 data products in order to classify the layers detected as either clouds or aerosols. The results obtained are used to assess the reliability of the cloud-aerosol discrimination (CAD) scores reported in the version 4.1 release of the CALIPSO data products. FKM is an unsupervised learning algorithm, whereas the CALIPSO operational CAD algorithm (COCA) takes a highly supervised approach. Despite these substantial computational and architectural differences, our statistical analyses show that the FKM classifications agree with the COCA classifications for more than 94 % of the cases in the troposphere. This high degree of similarity is achieved because the lidar-measured signatures of the majority of the clouds and the aerosols are naturally distinct and hence objective methods can independently and effectively separate the two classes in most cases. Classification differences most often occur in complex scenes (e.g., evaporating water cloud filaments embedded in dense aerosol) or when observing diffuse features that occur only intermittently (e.g., volcanic ash in the tropical tropopause layer). The two methods examined in this study establish overall classification correctness boundaries due to their differing algorithm uncertainties. In addition to comparing the outputs from the two algorithms, analysis of sampling, data training, performance measurements, fuzzy linear discriminants, defuzzification, error propagation, and key parameters in feature type discrimination with the FKM method are further discussed in order to better understand the utility and limits of the application of clustering algorithms to space lidar measurements. In general, we find that both FKM and COCA classification uncertainties are only minimally affected by noise in the CALIPSO measurements, though both algorithms can be challenged by especially complex scenes containing mixtures of discrete layer types. Our analysis results show that attenuated backscatter, and color ratio are the driving factors that separate water clouds from aerosols; backscatter intensity, depolarization, and mid-layer altitude are most useful in discriminating between aerosols and ice clouds; and the joint distribution of backscatter intensity and depolarization ratio is critically important for distinguishing ice clouds from water clouds.

## 1 Introduction

The Cloud-Aerosol Lidar and Infrared Pathfinder Satellite Observations (CALIPSO) mission has been developed through a close and on-going collaboration between NASA Langley Research Center (LaRC) and the French space agency, Centre National D'Etudes Spatial (CNES) (Winker et al., 2010). This mission provides unique measurements to improve our understanding of global radiative effects of clouds and aerosols in the Earth's climate system. The CALIPSO satellite was launched in April 2006, as a part of the A-Train constellation (Stephens and Vane, 2007). The availability of continuous, vertically resolved measurements of the Earth's atmosphere at global scale leads to great improvements in understanding both atmospheric observations and climate models (Konsta et al. 2013; Chepfer et al. 2008).

The Cloud-Aerosol Lidar with Orthogonal Polarization (CALIOP), on-board CALIPSO, is the first satellite-borne polarization-sensitive lidar that specifically measures the vertical distribution of clouds and aerosols along with their optical and geometrical properties. The level 1 CALIOP data products report vertically-resolved total atmospheric backscatter intensity at both 532 nm and 1064 nm, and the component of the 532 nm backscatter that is polarized perpendicular to the laser polarization plane. The level 2 cloud and aerosol products are retrieved from the level 1 data and separately stored into two different file types: the cloud, aerosol, and merged layer product files (CLay, ALay, and MLay, respectively) and the cloud and aerosol profile product files (CPro and APro). The profile data are generated at 5 km horizontal resolution for both clouds and aerosols, with vertical resolutions of 60 m from -0.5 km to 20.2 km, and 180 m from 20.2 km to 30 km. The layer data are generated at 5 km horizontal resolution for aerosols and at three different horizontal resolutions for clouds (1/3 km, 1 km and 5 km). The layer products consist of a sequence of column descriptors (e.g., latitude, longitude, time, etc.) that provide information about the vertical column of atmosphere being evaluated. Each set of column descriptors is associated with a variable number of layer descriptors that report the spatial and optical properties of each layer detected in the column.

The CALIOP level 2 processing system is composed of three modules, which have the general functions of detecting layers, classifying the layers, and performing extinction retrievals. These three modules are the Selective Iterated BoundarY Locator (SIBYL), the Scene Classifier Algorithms (SCA), and the Hybrid Extinction Retrieval Algorithms (HERA) (Winker et al. 2009). The level 2 lidar processing begins with the SIBYL module that operates on a sequence of scenes consisting of segments of level 1 data covering 80 km in along-track distance. The module averages these profiles to horizontal resolutions of 5, 20 and 80 km respectively, and detects features at each of these resolutions. Those features detected at 5 km are further inspected to determine if they can also be detected at finer spatial scales (Vaughan et al., 2009). The SCA is composed of three main sub-modules: the cloud and aerosol discrimination (CAD) algorithm (Liu et al., 2004, 2009, 2019), the aerosol subtyping algorithm (Omar et al., 2009; Kim et al., 2018), and the cloud ice-water phase discrimination algorithm (Hu et al., 2009; Avery et al., 2019). Profiles of particulate (i.e., cloud or aerosol) extinction and backscatter coefficients and estimates of layer optical depths are retrieved for all feature types by the HERA module.

Clouds and aerosols modulate the Earth's radiation balance in different ways, depending on their composition and spatial and temporal distributions, and thus being able to accurately discriminate between them using global satellite measurements is critical for better understanding trends in global climate change (Trenberth et al., 2009). The CALIOP operational CAD algorithm (COCA) uses a family of multi-dimensional probability density functions (PDFs) to distinguish between clouds and

aerosols (Liu et al., 2004, 2009, 2019). Using a larger number of layer attributes (i.e., higher dimension PDFs) generally yields increasingly accurate cloud and aerosol discrimination. While both V3 and V4 COCA algorithms use the same five attributes to derive their classifications, substantial improvements have been made in V4 due to much improved calibration, especially at 1064 nm (Liu et al, 2019; Vaughan et al., 2019). The V4 PDFs have been re-built to better discriminate dense dust over the Taklimakan desert, lofted dust over Siberia and the American Arctic regions, and high-altitude smoke and volcanic aerosol.

Also, the application of the V4 PDFs has been extended, and they are now used to discriminate between clouds and aerosols in the stratosphere and to features detected at single-shot resolute (333 m) in the mid-to-lower troposphere.

CALIPSO has been delivering separate cloud and aerosol data products throughout its 12+ year lifetime, and the reliable segregation of these products clearly depends on the accuracy of the COCA. However, to the best of our knowledge, no traditional validation study of the CALIOP CAD results has been published in the peer-reviewed literature. Traditional

validation studies typically compare coincident measurements of identical phenomena acquired by previously validated and well-established instruments to the measurements acquired by the instrument being validated. For example, radiometric calibration of the CALIOP attenuated backscatter profiles have been extensively validated using ground-based Raman lidars (Mamouri et al., 2009; Mona et al., 2009) and airborne high spectral resolution lidars (HSRL) (Kar et al., 2018; Getzewich et al., 2018). Furthermore, CALIOP level 2 products have also been thoroughly validated: cirrus cloud heights and extinction

coefficients have been validated using measurements by Raman lidars (Thorsen et al., 2011), Cloud Physics Lidar (CPL) measurements (Yorks et., 2011; Hlavka et al. 2012), and in situ observations (Mioche et al., 2010); CALIOP aerosol typing has been assessed by HSRL measurements (Burton et al., 2013) and Aerosol Robotic Network (AERONET) retrievals (Mielonen et al., 2009); and CALIOP aerosol optical depth estimates have been validated using HSRL measurements (Rogers et al., 2014), Raman measurements (Tesche et al., 2013), AERONET measurements (Schuster et al., 2012; Omar et al., 2013),

and Moderate Resolution Imaging Spectroradiometer (MODIS) retrievals (Redemann et al., 2012). These level 2 validation studies implicitly depend on the assumption that the COCA classifications are essentially correct; however, this fundamental assumption has yet to be verified. This paper is, therefore, a first step in an on-going process of verifying and validating the outputs of the CALIOP operational CAD algorithm. But unlike traditional validation studies in which coincident measurements are compared, this study will compare the outputs of two wholly different classification schemes applied to the same measured

input data. Clearly one of these two schemes is COCA. The other is the venerable fuzzy k-means (FKM) clustering algorithm, which has a long history of use in classifying features found in satellite imagery (Harr and Elsberry, 1995; Metternicht, 1999; Burrough et al., 2001; Olthof and Latifovic, 2007; Jabari and Zhang, 2013).

The rationale for comparing algorithm outputs rather than measurements is twofold. First, no suitable set of coincident observations is currently available for use in a global-scale validation study. The spatial and temporal coincidence of ground-based and airborne measurements is extremely limited, and thus any validation exercise would require assumptions about the compositional persistence of features being compared. (Paradoxically, these are precisely the sorts of assumptions that should

be obviated by well-designed validation studies.) Coincident A-Train measurements can be used in simple cases (Stubenrauch et al., 2013), but have little to offer in the complex scenes where cloud and aerosol intermingle; e.g., passive sensors cannot provide comparative information in multi-layer scenes or at cloud-aerosol boundaries, and the CloudSat radar is only sensitive to large particles, and thus cannot help to distinguish between scattering targets that it cannot detect (e.g., lofted dust and thin cirrus). Second, COCA is a highly supervised classification scheme whose decision-making prowess depends on human-

specified probability density functions (PDFs). FKM, on the other hand, is an unsupervised learning algorithm that, after suitable training, delivers classifications based on the inherent structure found in the data. The results obtained from the two different algorithms will help us better understand global cloud and aerosol distributions, which is important for all the users of space lidar (e.g., atmospheric scientists, weather and climate modelers, instrument developers, etc.). The flexibility of the FKM approach can help determine which individual parameters are most influential in discriminating clouds from aerosols

and help evaluate the degree of improvement to be expected if/when new observational dimensions are added to the COCA PDFs.

Our paper is structured as follows. Section 2 briefly reviews the fundamentals of the COCA PDFs and their application to the CALIOP measurements. Section 3 provides an overview of the FKM algorithm and describes how we have adapted it for use in the CALIOP cloud-aerosol discrimination task. Section 4 compares the FKM classifications to the V3 and V4 COCA

results. These comparisons, which are made for both individual cases and statistical aggregates, are designed to assess the accuracy of the COCA algorithm in general and to quantify changes in performance that can be attributed to the algorithm refinements incorporated in V4 (Liu et al., 2019). Various FKM performance metrics are described in Sect. 5, including error propagation, key parameter analysis, fuzzy discriminant analysis and principle component analysis. Conclusions and perspectives are given in Sect. 6.

**2 CALIOP CAD PDF construction**

The CALIOP operational CAD algorithm uses manually-derived, multi-dimensional PDFs together with a statistical discrimination function to distinguish between clouds and aerosols. Given a standard set of lidar measurements ($X_1$, $X_2$, … $X_m$), separate multidimensional PDFs are constructed for clouds ($P_{cloud}(X_1, X_2, … X_m)$) and aerosols ($P_{aerosol}(X_1, X_2, … X_m)$). Discrimination between clouds and aerosols for previously unclassified layers is then determined using

$$f\left(X_1, X_2, ..., X_m\right) = \frac{P_{cloud}\left(X_1, X_2, ..., X_m\right) - P_{aerosol}\left(X_1, X_2, ..., X_m\right)k}{P_{cloud}\left(X_1, X_2, ..., X_m\right) + P_{aerosol}\left(X_1, X_2, ..., X_m\right)k}. \tag{1}$$

The function f is a normalized differential probability, which value ranges from -1 to 1, and k is a scaling factor that is related to the ratio of the numbers of aerosol layers and cloud layers used to develop the PDFs (Liu et al. 2009; Liu et al., 2019). Within the CALIOP level 2 data products, a percentile (integer) value of $100 \times f$, ranging from -100 to 100, is reported as the "CAD score" characterizing each feature. Aerosol CAD scores range from -100 to 0, and cloud CAD scores range from 0 to 100. Because the nature of clouds is quite different from aerosols, most cloud and aerosols can be distinguished unambiguously. Transition regions where clouds are embedded in aerosols, volcanic ash injected into the upper troposphere, and optically thick, strongly-scattering aerosols at relatively high altitudes (e.g., haboobs) can still present significant discrimination challenges, but these cases occur relatively infrequently.

The initial version of COCA used only three layer attributes: layer mean attenuated backscatter at 532 nm, $<\beta'_{532}>$, layer-integrated attenuated backscatter color ratio, $\chi' = <\beta'_{1064}>/<\beta'_{532}>$, and mid-layer altitude, $z_{mid}$. Since then the algorithm has been incrementally improved, and beginning in V3 the COCA PDFs were expanded to five dimensions (5-D) by adding layer-integrated 532 nm volume depolarization, $\delta_v$, and the latitude of the horizontal mid-point of the layer (Liu et al., 2019). Within the CALIPSO analysis software these PDFs are implemented as 5-D arrays that function as look-up tables. However, while the V3 and V4 algorithms both use five independent measurements, the numerical values in the underlying PDFs are significantly different. The V3 PDFS were rendered obsolete by extensive changes to the V4 calibration algorithms (Kar et al., 2017; Getzewich et al., 2018; Vaughan et al., 2019) that required revising a number of the $<\beta'_{532}>$ probabilities and a global recalculation of the color ratio probabilities. As a result, the V4 CAD algorithm can now be applied in the stratosphere and to layers detected at single shot resolution and has greatly improved performance when identifying dense dust over the Taklimakan desert, lofted dust over Siberian and American Arctic regions, and high-altitude aerosols in the upper troposphere and lower stratosphere (Liu et al., 2019).

## 3 Fuzzy k-means cluster analysis

Cluster analysis is a useful statistical tool to group data into several categories and has been successfully applied to satellite observations to discriminate among different features of interest (Key et al., 1989; Kubat et al. 1998; Omar et al., 2005; Zhang et al., 2007; Usman, 2013; Luo et al., 2017; Gharibzadeh et al., 2018). There are many different types of clustering methods, such as connectivity-based, centroid-based, density-based, and distribution clustering, and these are typically trained using either supervised or unsupervised learning techniques. In this paper, we focus on a centroid-based, unsupervised learning approach known as the fuzzy k-means (FKM) method. As the name implies, classification ambiguities are expressed in terms

of fuzzy logic (i.e., as opposed to "crisp"/binary logic) and thus every point processed by the clustering algorithm is assigned some degree of membership in all categories, rather than belonging solely to just one category.  FKM membership values range from 0 to 1, and thus are comparable to the operational CAD scores.  In addition, the shapes and density distributions of multi-dimensional observations of clouds and aerosols from lidar well-suited for the centroid-based clustering technique used

by the FKM classification method. With the exception of latitude, our FKM implementation uses the same inputs as COCA; i.e., $<\beta'_{532}>$, $\chi'$, $\delta_v$, and $z_{mid}$.  We make this choice because clouds and aerosols show distinct centers in the $<\beta'_{532}>$, $\delta_v$, $\chi'$, $z_{mid}$ attribute space, whereas adding latitude degrades the separation between cluster centers and adds significantly to class overlap. A key parameter analysis (described in Sect. 5) demonstrates that latitude does not provide intrinsic information that helps to distinguish between aerosols and cloud, nor does it improve the reliability of the cluster membership values (e.g., Wilks'

lambda, a measure of the difference between classes also introduced in Sect 5., deteriorates from ~0.2 to ~0.5).  However, for probabilistic systems (e.g., COCA) latitude can be useful, simply because some feature types are more likely than others to occur within specific latitude-altitude bands (e.g., at altitudes of 9–11 km, significant aerosol loading is much more likely at 45° N than at 60° S).

## 3.1 FKM algorithm architecture

Given a set of observations X = (X$_1$, X$_2$, …, X$_n$), where each observation is a *p*-dimensional real vector, FKM logical clustering aims to partition the n observations into k (≤ n) sets S = {S$_1$, S$_2$, …, S$_k$} so as to "minimize the within-cluster sum of squares (WCSS) and maximize the between cluster sum of squares (BCSS)" (Hartigan and Wang 1979). Points on the edge of a cluster may be in the cluster to have a lesser degree than points in the center of the cluster. The clustering results (i.e., fuzzy memberships, organized into a matrix, M, with elements m$_{ij}$, i = 1…n;  j = 1...k) are assigned values between 0 and 1 (Eq. 2).

When elements of the membership matrix, m=1, an individual i belongs only to a single class j and has a class membership of 0 in all other classes. Note also that in the standard (i.e., not fuzzy) k-means algorithm m$_{ij}$ can be only 1 or 0 (i.e., a point can only belong to one cluster), but that intermediate values are permitted in the FKM method (i.e., a point can partially belong to a particular cluster). The sum of the fuzzy memberships for an individual over all classes is equal to one (Eq. 3), and there will be at least one individual with some non-zero membership belonging to each class (Eq. 4). These defining relationships are

written as

$$m_{ij} \in [0,1], i = 1...n, j = 1...k \tag{2}$$

$$\sum_{j=1}^{k} m_{ij} = 1, i = 1...n \text{ , and} \tag{3}$$

$$\sum_{i=1}^{n} m_{ij} > 0, \; j = 1...k \; . \tag{4}$$

To determine the best solution, based on minimization of the WCSS, a classic objective function, J, is built so that the best solution is the one that minimizes *J* (Bezdek, 1981; Bezdek, 1984; McBratney and Moore, 1985). The functional form of J is

$$J(M,C) = \sum_{i=1}^{n} \sum_{j=1}^{k} m_{ij}^{\phi} d_{ij}^{2} \left( x_{il}, c_{jl} \right), \tag{5}$$

where C ($c_{jl}$; j=1,.., k; l =1,..., p) is a matrix of class centers, and $d^2(x_{il}, c_{jl})$ is the squared distance between individual $x_{il}$ and class center $c_{jl}$ according to a chosen definition of distance (e.g., the Mahalanobis distance; see Sect. 2.3). The objective function is the squared error from class centers weighted by the $\phi^{th}$ power (fuzzy weighting exponent) of the membership values. For the least meaningful value, $\phi = 1$, J minimizes only at crisp partitions (the memberships converge to either 0 or 1), with no overlap between cluster boundaries. Increasing the value of $\phi$ tends to degrade memberships towards fuzzier states

where there are more overlaps between the boundaries of clusters. For a specified value of $\phi$, minimization of objective function J optimizes the solutions for the membership matrix M and its associated centroid matrix C (Bezdek, 1981; McBratney and deGruijter, 1992; Minasny and McBratney, 2002). Class centers are the averages of the individual samples weighted by their class membership values raised to the $\phi^{th}$ power (Eq. 6). The membership ($m_{ij}$) of an individual belonging to a class is the distance between the individual and the class center divided by the sum of the distances between the individual and the centers

of all classes (Eq. 7), or

$$c_{jl} = \frac{\sum_{j=1}^{n} m_{ij}^{\phi} x_{ij}}{\sum_{i=1}^{n} m_{ij}^{\phi}}, \; j = 1, 2...k, \; l = 1, 2...p \; , \; \text{and} \tag{6}$$

$$m_{jl} = \frac{d_{ij}^{-2/(\phi-1)}}{\sum_{j=1}^{k} d_{ij}^{-2/(\phi-1)}}, \; i = 1, 2...n, \; j = 1, 2...k \; , \; \text{and} \tag{7}$$

To obtain centroid (Eq. 6) and membership (Eq. 7) solutions, Picard iterations (Bezdek et al., 1984) are applied until the centers or memberships are constant to within some small value (see the algorithm flowchart in Fig. 1). We first initialize the

memberships as random values using a uniform distribution that satisfies all conditions given by equations 2, 3 and 4. We then calculate class centers and recalculate memberships according to the new centers. If the new memberships do not change

compared to the old ones (or change only within a small difference $\varepsilon$), the clustering process ends. Otherwise we recalculate the new centers and new memberships. If the algorithm does not converge after a fixed number of iterations, the procedure is reinitiated using newly (and again randomly) specified initial cluster centers. This process repeats until the algorithm converges to a point where the relative change in the objective function (calculated from Eq. 5, which quantifies the changes in both the memberships and centers) is less than $\varepsilon$ (0.001) and saves the best memberships and centers that result from the optimum random initiation corresponding to the least objective function.

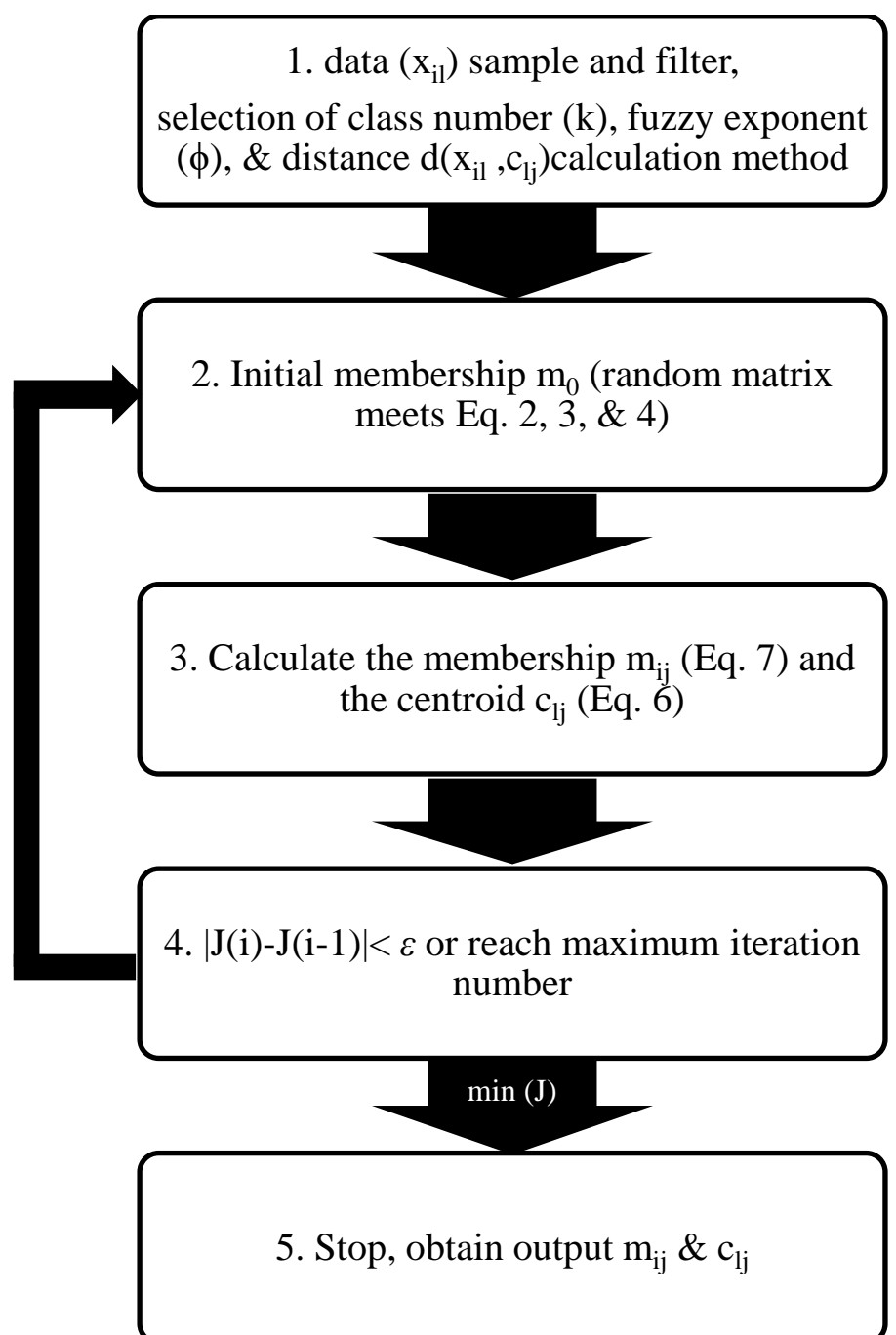

Figure 1: Flowchart illustrating the operation of the Fuzzy K-means algorithm

Before running the FKM code (from Minasny and McBratney, 2002), we prepared our data by sampling, training and filtering (Sect. 2.1 and Sect. 2.2). We also selected a reasonable method to calculate the distance between individuals and centers (Sect 2.3) and determined optimal values for class number and fuzzy exponent (Sect. 2.4). Note the FKM method is directly applied to data to get membership instead of building PDF as in operational algorithm.

**3.2 Data sample and training**

As mentioned above, four level 2 parameters are used for our cluster analysis: layer mean attenuated backscatter at 532 nm, $<\beta'_{532}>$, layer integrated volume depolarization ratio at 532 nm, $\delta_v$, total attenuated backscatter color ratio, $\chi'$, and mid-layer altitude, $z_{mid}$. The selection of four dimensions is based on many previous studies (e.g., Liu et al., 2004, 2009; Hu et al., 2009; Omar et al., 2009; Burton et al., 2013), which show that clouds, aerosols and their subtypes are quite different based on these

observations. $<\beta'_{532}>$, $\delta_v$, and $\chi'$ are the fundamental lidar-derived optical properties that form the basis for our discrimination scheme. We also include altitude, as the joint distributions of altitude with the various lidar optical properties have proved to be highly effective in identifying different feature types.

In this study we apply FKM at a global scale. For any given region, results derived from a localized cluster analysis will likely give us better classifications compared to the results from a global scale analysis, but investigating and/or characterizing these

differences lies well beyond the scope of this study. The data sample size also strongly influences the clustering results. For example, clustering into two classes with a full complement of CALIPSO data could identify clear and "not clear" scenes. If clear scenes are excluded, clustering could separate clouds and aerosols. If only clear scenes are included, clustering could possibly provide a means of identifying different surface types. With only cloudy data, clustering could be used to derive thermodynamic phase classification. With only aerosol data, clustering is actually aerosol subtyping. With only liquid cloud

data, clustering could separate cumulus and stratocumulus. So, the size and composition of the dataset is very important for our analysis, which strongly depends on the objective of the classification.

To extrapolate the classification of identifiable elements using FKM from a small subset to a broader population, we identify an appropriate training data set from which the classifications can be derived (Burrough et al., 2000). This training data should be representative of the broader sample for which the classification will be implemented (i.e., both must span similar domains).

To ensure the selection of an appropriate training data set, the shapes of the PDFs of the relevant parameters derived from any proposed training set should closely match the shapes of the corresponding parameter PDFs derived from the global long-term data set. Data from the month of January 2008 is used to determine the optimal number of classes ($k$) and fuzzy exponent ($\phi$) required for classification and optimal values of the performance parameters, and to calculate class centroids for interpretation of similarities and differences between classes. To avoid errors due to small sample sizes, we used the same month of global

observations (January 2008) to do the subsequent comparisons with COCA results.

Figure 2 shows approximate probability density functions (computed by normalizing the sum of the occurrence frequencies to 1) for different lidar observables for liquid water clouds, randomly- and horizontal-oriented ice clouds, and aerosols during all of January 2008. Liquid water clouds have the largest $<\beta'_{532}>$ and $\chi'$ values compared with other species. Aerosols generally have the smallest $\chi'$, $\delta_v$, and $<\beta'_{532}>$, and ice clouds have the largest $\delta_v$ compared with the other two species. There is overlap between species, but these three parameters are still sufficient to separate aerosols and different phases of cloud in most cases. The three bottom panels (d, e, & f) in Figure 2 are from a single half orbit (2008-09-06T01-35-29ZN) of observations. The PDFs of one-half orbit and one month of observations appear to agree very well, which means that focused feature clustering studies that use the FKM method can be applied to a small sample such as one-half orbit of observations and not cause significant biases to the standard full dataset.

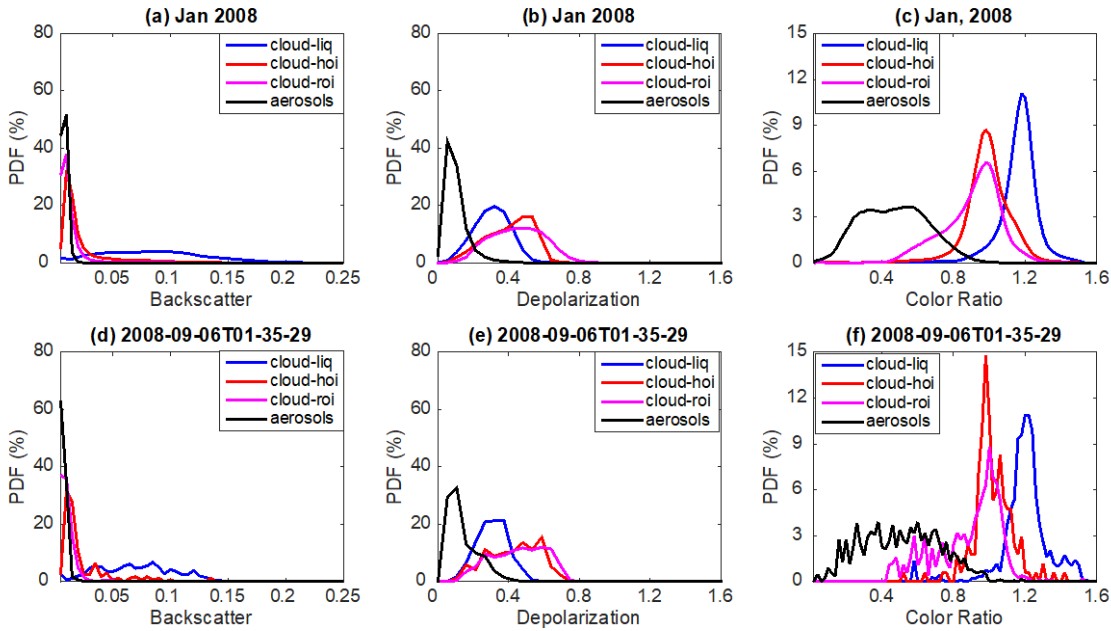

Figure 2: Comparisons of approximate probability density functions computed by normalizing the sum of the occurrence frequencies to 1; top row (panels a – c) shows data from all of January 2008; bottom row (panels d – f) shows data from a single half-orbit (06 Sep. 2008, 01:35:29 GMT). The left column (panels a and d) compares total attenuated backscatter PDFs; the center column (panels b and e) compares volume depolarization ratio PDFs; and the right column (panels c and f) compares total attenuated backscatter color ratio PDFs. Black lines represent aerosols, blue lines represent liquid water clouds, red lines represent ice clouds dominated by horizontal oriented ice (HOI), and magenta lines represent ice clouds dominated by random oriented ice (ROI).

### 3.3 Data filtering

We filtered the training data to eliminate outliers in the $\langle\beta'_{532}\rangle$, $\delta_v$, and $\chi'$ measurements that were physically unrealistic (i.e., either too high or too low). Eliminating these extreme values speeds up the processing, and the training algorithm converges more rapidly. The selected filter thresholds retain more than 98 % of all features within the original data set. A summary of the thresholds is given in Table 1. The selection of these thresholds is based on the PDFs shown in Figure 2.

**Table 1: Filter thresholds for FKM lidar observables**

| Lidar observable | Filter criteria |
|---|---|
| Mean attenuated backscatter at 532 nm | $0 \leq \langle\beta'_{532}\rangle \leq 0.2 \text{ sr}^{-1} \text{ km}^{-1}$ |
| Integrated volume depolarization ratio | $0 \leq \delta_v \leq 2$ |
| Total attenuated backscatter color ratio | $0 \leq \chi' \leq 2$ |

### 3.4 Distance calculation

The distances between attributes can be calculated in different ways (e.g., Euclidean distance, Diagonal distance and Mahalanobis distance). According to a study by Gorsevski et al. (2003), we should apply the Euclidean distance to uncorrelated variables on the same scale when attributes are independent and the clusters are spherically shaped clouds. The Diagonal distance is also insensitive to statistically-dependent variables but clusters are not required to have spherically-shaped clouds. The Mahalanobis distance can be used for correlated variables on the same or different scales and when the clusters are ellipsoidal-shape clouds. The Mahalanobis distance ($d_{ij}$) of an observation i from a set of observations ($x_{il}$) with centers $c_{jl}$ ($x_{il}$ - $c_{jl}$ is an l-dimensional vector) is defined in Eq. 8 (Mahalanobis, 1936):

$$d_{ij}^2 = \left(x_{il} - c_{jl}\right)^T S^{-1} \left(x_{il} - c_{jl}\right), \; i = 1, 2...n, \; j = 1, 2...k, l = 1, 2...p \; . \tag{8}$$

$S^{-1}$ (an l×l matrix) is the inverse of the covariance matrix of the observations. Note superscript $T$ indicates that the vector should be transposed. If covariance matrix is a diagonal matrix, the Mahalanobis distance calculation returns the normalized Euclidean distance. In this work we use the Mahalanobis distance specifically because the three lidar observables used both in FKM and COCA are not independent. Each is a sum (or mean) of the measured backscatter signal over some altitude range, with the relationships between them given as follows:

$$\langle\beta'_{532}\rangle = \frac{1}{N} \sum_{n=1}^{N} \beta'_{532,\parallel}\left(z_n\right) + \beta'_{532,\perp}\left(z_n\right), \tag{9a}$$

$$\delta_v = \left. \sum_{n=1}^{N} \beta'_{532,\perp}(z_n) \middle/ \sum_{n=1}^{N} \beta'_{532,\parallel}(z_n) \right. ,$$

(9b)

$$\text{and} \quad \chi' = \left\langle \beta'_{1064} \right\rangle \middle/ \left\langle \beta'_{532} \right\rangle = \left. \sum_{n=1}^{N} \beta'_{1064}(z_n) \middle/ \sum_{n=1}^{N} \beta'_{532}(z_n) \right. .$$

(9c)

In these expressions, the subscripts $\parallel$ and $\perp$ represent contributions from the 532 nm parallel and perpendicular channels, respectively. Note in particular that the signals measured in the 532 nm parallel channel contribute to all three quantities.

## 3.5 The choice of class *k* and fuzzy exponent *ϕ*

The selection of an optimal number of classes k ($1 < k < n$) and degree of fuzziness $\phi$ ($\phi > 1$) has been discussed in many previous studies (Bezdek, 1981; Roubens, 1982; McBratney and Moore, 1985; Gorsevski, 2003). The number of classes specified should be meaningful in reality and the partitioning of each class should be stable. For each generated classification,

analyses need to be performed to validate the results. Among different validation functions, the fuzzy performance index (FPI) and the modified partition entropy (MPE) are considered two of the most useful indices among seven examined by Roubens (1982) to evaluate the effects of varying class number. The FPI is defined as in Eq. 10, where *F* is the partition coefficient calculated from Eq. 11. The MPE is defined as in Eq. 12, with the entropy function (H) calculated from Eq. 13.

The ideal number of continuous and structured classes (k) can be established by simultaneously minimizing both FPI and

MPE. For the fuzziness exponent, if the value of $\phi$ is too low the classes become more discrete and the membership values either approach 0 or 1. But if $\phi$ is too high, the classes will not provide useful discrimination among samples and classification calculations may fail to converge. McBratney and Moore (1985) suggested that the objective function (Eq. 14, Bezdek, 1981) decreases with increasing of both fuzzy exponent ($\phi$) and the number of classes (k). They plotted a series of objective functions versus the fuzzy exponent ($\phi$) for a given class where the best value of $\phi$ for that class is at the first maximum of objective

function curves (Odeh et al. 1992a, McBratney and Moore 1985). Therefore, choosing an optimal combination of class number (k) and fuzzy exponent ($\phi$) is established on the basis of minimizing both values of FPI and MPE and the least maximum of the objective function, where

$$\text{FPI} = 1 - \frac{k \times F - 1}{k - 1}, \text{ and} \tag{10}$$

$$F = \frac{1}{n} \sum_{i=1}^{n} \sum_{j=1}^{k} m_{ij}^{2}; \tag{11}$$

$$\text{MPE} = \frac{H}{\log k}, \text{ and} \tag{12}$$

$$H = \frac{1}{n} \sum_{i=1}^{n} \sum_{j=1}^{k} m_{ij} \times \log\left(m_{ij}\right); \text{ and} \tag{13}$$

$$\frac{\partial J(M,C)}{\partial \phi} = \sum_{i=1}^{n} \sum_{j=1}^{k} m_{ij}^{\phi} \log\left(m_{ij}\right) d_{ij}^{2}. \tag{14}$$

Using one month of layer optical properties reported in the CALIOP level 2 merged layer products, we created Figure 3 to determine optimal values for k and $\phi$. From this figure, we conclude that the ideal number classes for CALIOP layer classification is either 3 or 4, with corresponding fuzzy exponents equal to 1.4 or 1.6 (we use 1.4 for the analyses this paper). Before exploring the clustering results to see what each class represents, we can immediately confirm that using three classes would be physically meaningful (i.e., these 3 classes may be aerosols, liquid water clouds and ice clouds). Similarly, two classes could represent aerosols and clouds. In the following study, we will choose k equal to 2 or 3 and $\phi$ equal to 1.4.

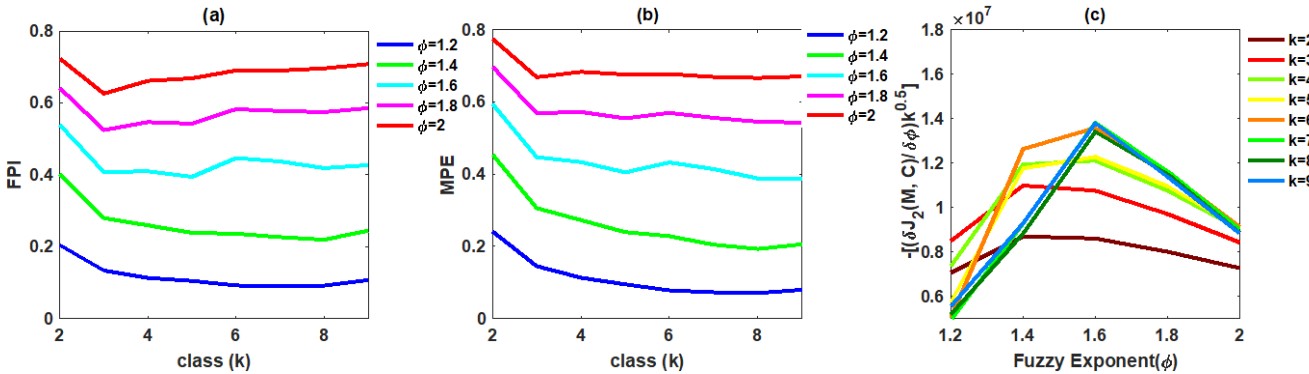

Figure 3: Determination of the number of classes, k, and the fuzzy exponent, $\phi$, for the FKM cloud-aerosol discrimination algorithm: (a) FPI (y-axis) versus class number *k* (x-axis) for different values of fuzzy exponent $\phi$ (different colors); (b) MPE (y-axis) versus class number k (x-axis) for different values of fuzzy exponent $\phi$ (different colors); and (c) objective function

values (y-axis) versus the fuzzy exponent ϕ (x-axis) for various class numbers (different colors).

## 4 Cluster results and comparison with V3 and V4 data

### 4.1 CAD from the Fuzzy K-Means algorithm

According to Liu et al. (2009), the CAD score for any layer is the difference between the probability of being a cloud and the probability of being an aerosol (Eq. 15). We calculate the FKM CAD score in a similar way, where the COCA probabilities are replaced with FKM membership values. For the 3-class FKM analyses, the cloud membership value is the sum of memberships of ice and water clouds (two classes). The FKM CAD score is found using

$$CAD_{FKM} = \frac{M_{cloud} - M_{aerosol}}{M_{cloud} + M_{aerosol}} \times 100 \,. \tag{15}$$

Figure 4 compares the operational V3 and V4 CAD products and our CAD$_{FKM}$ classifications for a single nighttime orbit segment (06 September 2008, beginning at 01:35:29 GMT). Generally speaking, CAD$_{FKM}$ from both the 2-class and 3-class analyses are quite similar to both the V3 and V4 COCA values. When COCA CAD scores are positive (namely clouds, shown in whitish colors in Fig. 4) in V3 and V4, the 2-class and 3-class CAD$_{FKM}$ values are also positive. Likewise, when COCA CAD scores are negative (namely aerosols, yellowish colors in Fig. 4), the 2-class and 3-class CAD$_{FKM}$ values are also negative. Furthermore, the particular orbit selected here includes the observations of a plume of high, dense smoke lofted over low water clouds (latitudes between 0° and 20° S, shown within the red oval in Fig. 4a). For these water clouds beneath dense smoke, both the V3 operational CAD and the 2-class CAD$_{FKM}$ label them as clouds with low positive values. On the other hand, the V4 operational CAD and the 3-class CAD$_{FKM}$ return higher values much closer to 100. The reasons for these differences will be discussed in Sect. 5.2 and Sect. 5.4. Note too that weakly scattering edges of cirrus clouds (hereafter, cirrus fringes) beyond 69.6° S are misclassified as aerosols by both the 2-class and 3-class CAD$_{FKM}$ (Figure 4c and d) but are correctly classified as cloud by the operational V4 algorithms.

The differences between the FKM and V4 COCA classifications are most prominent for layers detected in the stratosphere (i.e., layers rendered in black in Fig. 4b). In the northern hemisphere, between 71.55° N and 36.86° N, a diffuse, weakly scattering layer is intermittently detected at altitudes between 12 km and 18 km. This layer most likely originated with the eruption of the Kasatochi volcano on 7 August 2008 (Krotkov et al., 2010). But while V4 COCA classifies these layers as aerosols, both FKM methods identify them as clouds. In the southern hemisphere south of 69.6° S, a faint polar stratospheric layer is detected continuously, with a mean base altitude at ~ 11 km and a mean top at ~ 15 km. Once again, the V4 COCA classifies this feature as a moderate-to-high confidence aerosol layer and the 3-class FKM classifies it as a high confidence cloud. However, unlike the northern hemisphere case, the 2-class FKM identifies this as low confidence aerosol. Correctly

classifying stratospheric features that occupy the "twilight zone" between aerosols and highly tenuous clouds (Koren et al., 2007) is likely to be difficult for unsupervised learning methods, due to the extensive overlap in the available lidar observables for the two classes. Class separation is typically (though not always) more distinct within the troposphere.

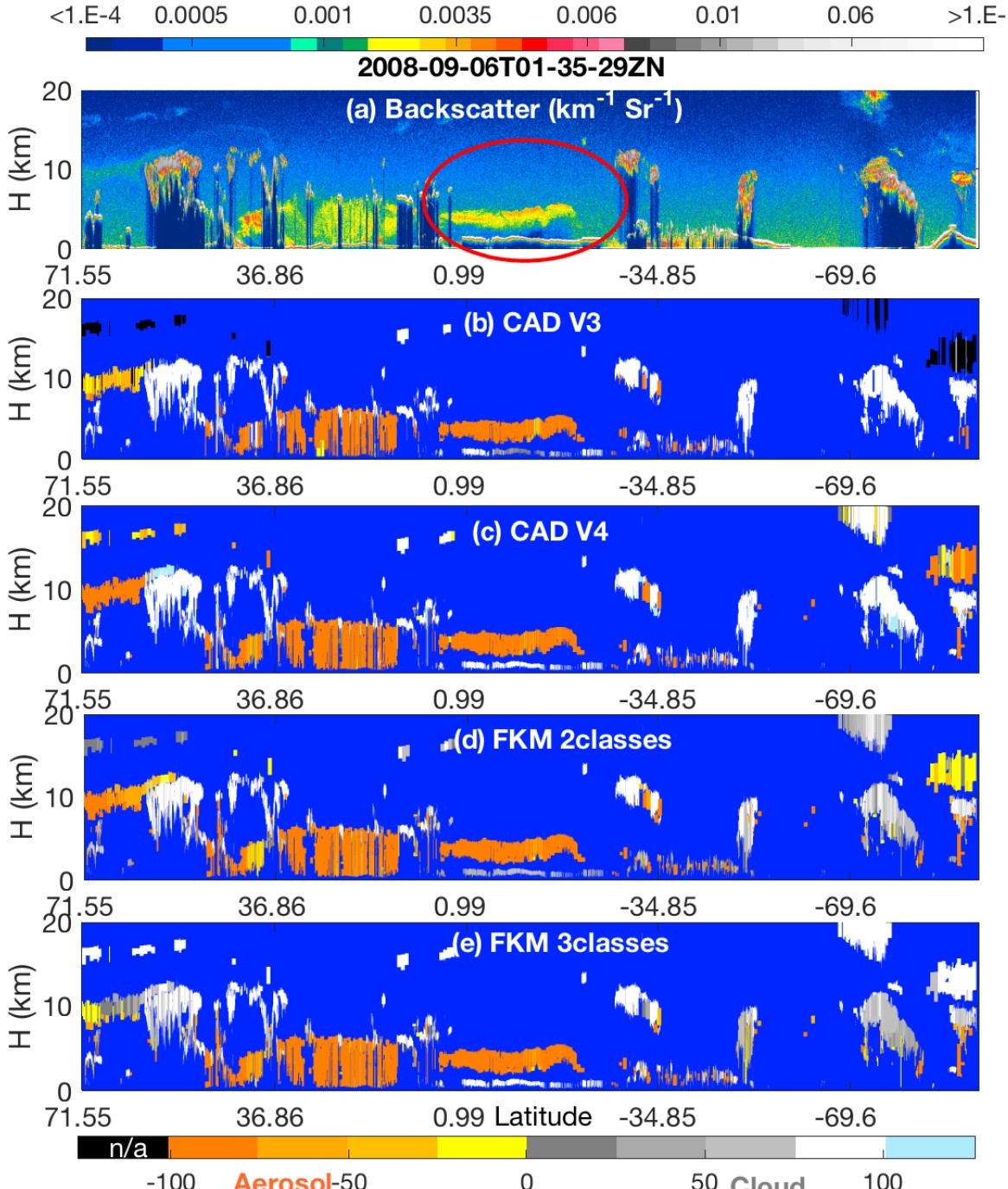

Figure 4: nighttime orbit segment from 6 September 2008, beginning at 01:35:29 UTC. The upper panel (a) shows 532 nm attenuated backscatter coefficients. The panels below show the CAD results as determined by (b) the V3 operational CAD algorithm, (c) the V4 operational CAD algorithm, (d) the 2-class FKM CAD algorithm, and (e) the 3-class FKM CAD algorithm. The red ellipse in the upper panel highlights a dense smoke layer lying above an opaque stratus deck. In the CAD images (panels b–e), stratospheric layers are shown in black, cirrus fringes are shown in pale blue, and regions of "clear air" where no features were detected are shown in pure blue. Latitude units in degrees; positive: north, negative: south.

## 4.2 Uncertainties: class overlap

The confusion index (CI) is a measure of the degree of class overlap or uncertainty between classes (Burrough and McDonnell, 1998). In effect, it measures how confidently each individual observation has been classified. CI values are calculated from Eq. 16, where $m_{max}$ denotes the biggest membership value and $m_{max-1}$ is the second biggest membership value for each individual observation (i):

$$CI = \left[ 1 - \left( m_{max_i} - m_{(max-1)_i} \right) \right].$$ (16)

CI value approaches zero when $m_{max}$ is much larger than $m_{max-1}$, indicating that the observation is more likely to belong to one dominant class. CI approaches one when $m_{max}$ is almost equal to $m_{max-1}$. In such cases, the difference between the dominant and subdominant classes is negligible, which creates confusion in the classification of that particular observation. Note the value $(1- CI) \times 100$ for the 2-class FKM algorithm is equivalent to the absolute value of the $CAD_{FKM}$ score.

Figure 5 shows CI values for 2-class and 3-class $CAD_{FKM}$ calculated for all layers in the sample orbit. From the figure, we see that, in most cases, the CI values are low for both the 2-class and 3-class $CAD_{FKM}$ classifications. The exceptions are stratospheric features (mostly near polar regions), cloud fringes, high altitude aerosols and, for 2-class $CAD_{FKM}$ only, the liquid water clouds beneath dense smoke. Low CI values for the $CAD_{FKM}$ classifications are analogous to high CAD scores assigned by the operational CAD algorithm: both indicate high confidence classifications. Similarly, $CAD_{FKM}$ classifications with high CI values indicate low confidence classifications where the observation has roughly equal membership in two classes. For the liquid water clouds beneath dense smoke, the membership values determined by the 2-class $CAD_{FKM}$ are larger than 0.5. However, the 3-class $CAD_{FKM}$ results for these water clouds have low CI values, indicating high confidence classifications into one dominant class, and suggesting that the separation between the aerosols and low water clouds is better accomplished when 3 classes are used. For cloud fringes, the CI values are high for both the 2-class and 3-class $CAD_{FKM}$. According to the $CAD_{FKM}$ results, cirrus fringes are somewhat different from the neighboring portions of the cirrus layer, as they also bear some similarity to the dust particles that are the predominant sources of ice nuclei (DeMott et al., 2010).

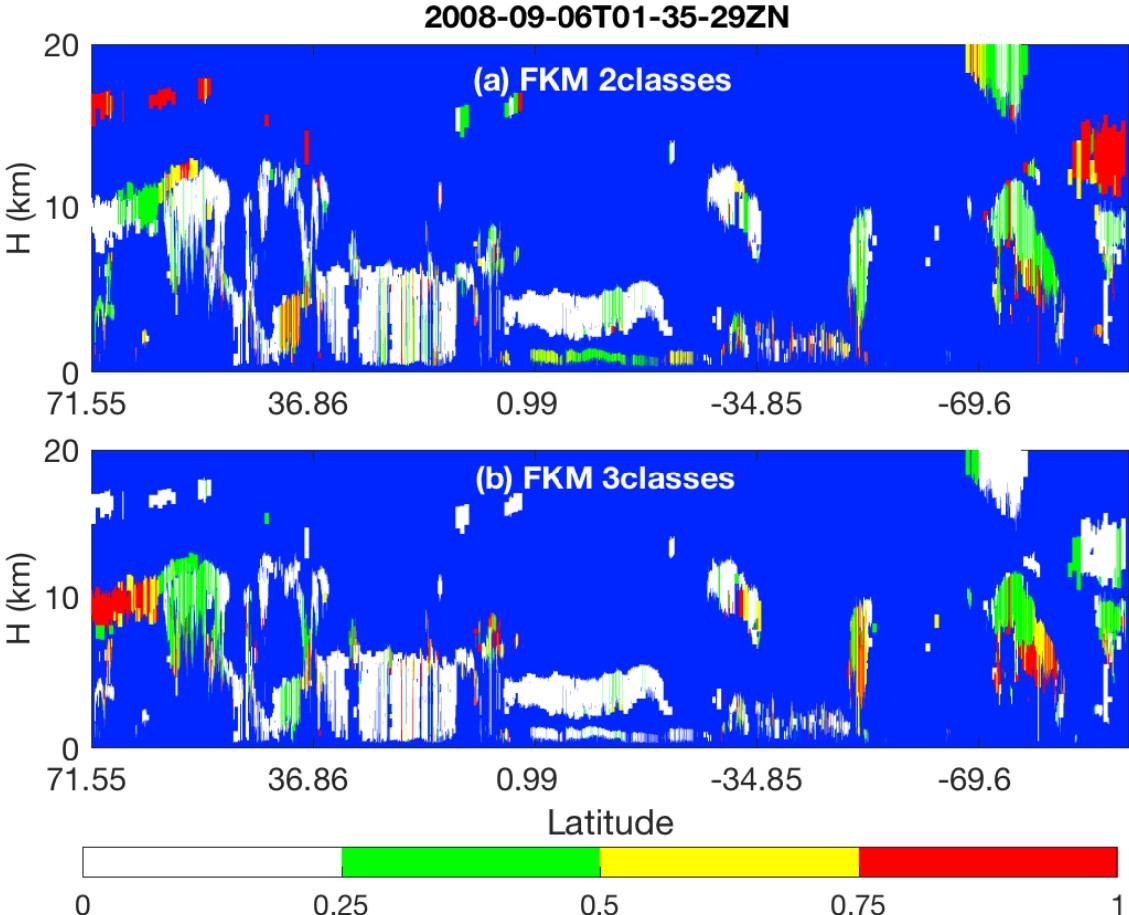

Figure 5: for the same data shown in Figure 4, the upper panel (a) shows the confusion index for 2-class $CAD_{FKM}$, and the lower panel (b) shows the confusion index for 3-class $CAD_{FKM}$. The pure blue color once again indicates those regions where no atmospheric layers were detected.

**4.3 Statistical comparisons of clouds and aerosols**

In this sub-section, we present statistical analyses of our results for all of January 2008, followed by explorations of individual case studies in the next sub-section. We first compare the PDFs of the different lidar optical parameters used in the 2-class and 3-class $CAD_{FKM}$ classifications to the PDFs of those same parameters derived for the COCA classifications (Fig. 6). We also compare the spatial distribution patterns of the clouds and aerosols identified by FKM and COCA (Fig. 7) and use confusion matrices to quantify the similarity of the corresponding FKM and COCA classes (Table 2).

From Fig. 6, it is evident that the PDFs of $<\beta'_{532}>$, $\delta_v$, and $\chi'$ that characterize the clouds and aerosols determined by the FKM classifications agree well with the PDFs from the V4 CAD classifications. Figures 6d, 6e, and 6f compare the 2-class $CAD_{FKM}$ results to the operational algorithm. In these figures, the PDFs of $<\beta'_{532}>$ (Fig. 6d), $\delta_v$ (Fig. 6e) and $\chi'$ (Fig. 6f) of FKM class 1 (blue dashed lines) agree well with those of V4 cloud PDFs (blue solid lines), while the PDFs of these different parameters of FKM class 2 (red dashed lines) agree well with those of V4 aerosol (red solid lines) PDFs. Figures 6a, 6b, and 6c compare the 3-class $CAD_{FKM}$ results to the operational algorithm. Once again, the comparisons are quite good: the shapes of the PDFs of FKM class 1 (blue dashed) agree well with the V4 water cloud (blue solid) PDFs, while the PDFs of FKM class 2 (red dashed) and 3 (green dashed) individually agree well with, respectively, the V4 ice cloud (red solid) and aerosol (green solid) PDFs. The class means for $<\beta'_{532}>$ are smallest for aerosols/class 3 (0.0034 ± 0.0022 (km$^{-1}$sr$^{-1}$) and 0.0041 ± 0.0193 (km$^{-1}$sr$^{-1}$), respectively) and slightly larger for ice clouds/ class 2 (0.0075 ± 0.0086 (km$^{-1}$sr$^{-1}$) and 0.0062 ± 0.0183 (km$^{-1}$sr$^{-1}$), respectively). Water clouds/class 1 have the largest $<\beta'_{532}>$ mean values (0.0804 ± 0.0526 (km$^{-1}$sr$^{-1}$) and 0.0850 ± 0.0454 (km$^{-1}$sr$^{-1}$), respectively). For $\delta_v$, the largest mean values are found for ice clouds/ class 2, followed by water clouds/class 1 and then aerosol/class 3. Class mean $\chi'$ is largest for water clouds/class 1 and smallest for aerosols/class 3. These means and standard deviations are also comparable between COCA and FKM classes.

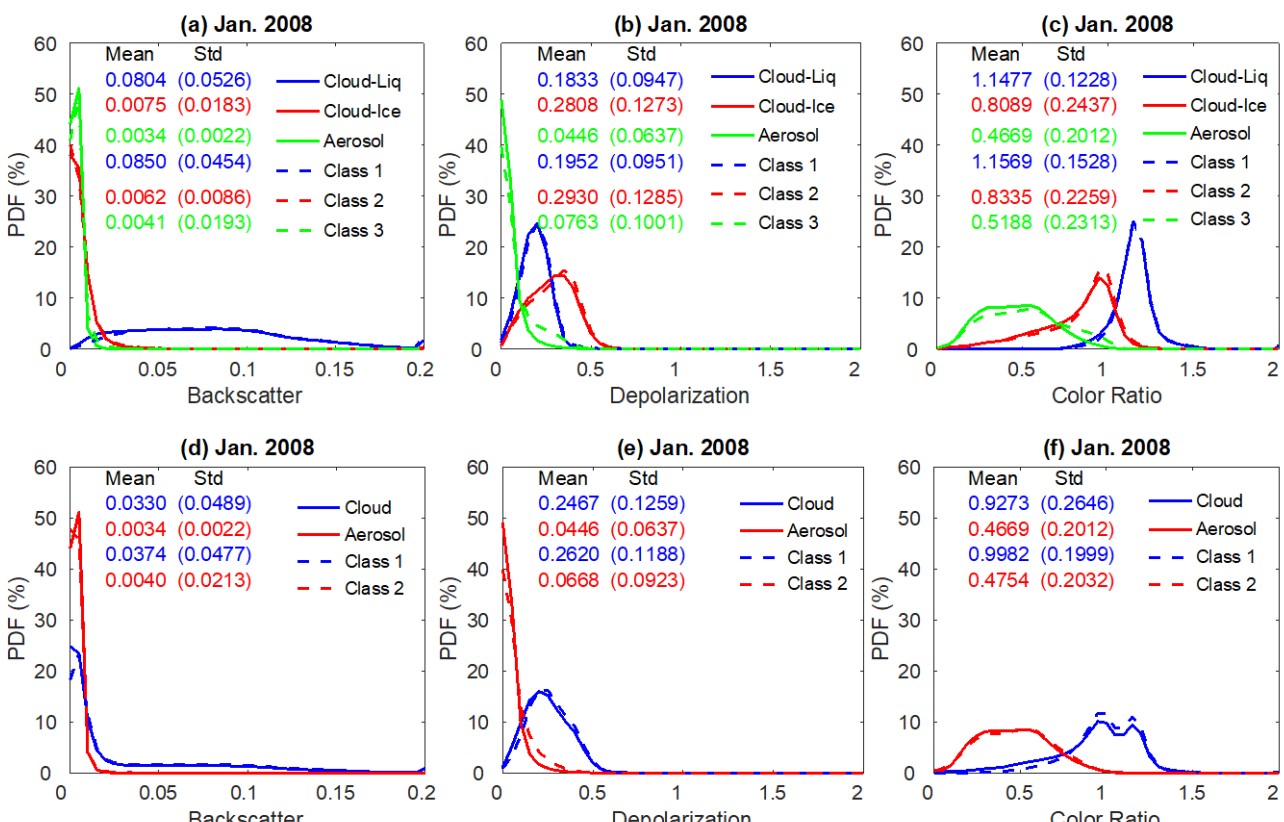

Figure 6: PDFs derived from all data from January 2008. The top row compares V4 operational CAD PDFs to the PDFs derived from CAD$_{FKM}$ 3-class results. V4 CAD PDFs for liquid water clouds, ice clouds, and aerosols are plotted in, respectively, solid blue, red and green lines. Similarly, CAD$_{FKM}$ 3-class PDFs for classes 1, 2, and 3 are plotted in, respectively, dashed blue, red and green lines. The bottom row compares V4 operational CAD PDFs to the PDFs derived from CAD$_{FKM}$ 2-class results, where once again the V4 CAD PDFs are shown in solid lines and the CAD$_{FKM}$ 2-class PDFs are shown in dashed lines. PDFs of $<\beta'_{532}>$ are shown in the left column (panels a and d), $\delta_v$ PDFs in the center column (panels b and e), and $\chi'$ PDFs in the right column (panels c and f).

Figure 7 compares the geographical (panels a-f) and zonally-averaged (panels g-l) distributions of 2-class CAD$_{FKM}$ occurrence frequencies to the COCA cloud and aerosol occurrence frequencies for all data acquired during January 2008. The spatial distributions of clouds and aerosols are quite different. In January, clouds are mostly located in the storm tracks, to the east of continents, over the inter-tropical convergence zone (ITCZ) and in polar regions. Aerosols are more often found over the Sahara, over the subtropical oceans, and in south-central and east Asia (upper two rows of Fig. 7). In the zonal mean plots (lower two rows of Fig. 7), cloud tops are seen to extend up to the local tropopause, whereas aerosols are largely confined to

the boundary layer. The geographical and vertical distributions of FKM class 1 are quite similar to the COCA V4 cloud distributions. Likewise, the distributions of FKM class 2 closely resemble the COCA V4 aerosol distributions. Looking at the difference plots (right-hand column of Fig. 7), some fairly large differences are seen in the polar regions, where the composition and intermingling of clouds and aerosols is notably different from other regions of the globe. Many of the layers observed in

the polar regions are spatially diffuse and optically thin, and thus occupy the morphological twilight zone between clouds and aerosols (Koren et al., 2007). Observationally-based validation of the feature types in these regions would likely require extensive in situ measurements coincident with CALIPSO observations. Consequently, correctly interpreting the classifications by the two algorithms in polar regions based on our knowledge is too challenging to draw useful conclusions and lies well beyond the scope of this work. Nevertheless, the PDFs and geographic analyses presented here establish that,

excluding the polar regions, the cloud-aerosol discrimination derived using an unsupervised FKM method is statistically consistent with the classifications produced by the operational V4 CAD algorithm.

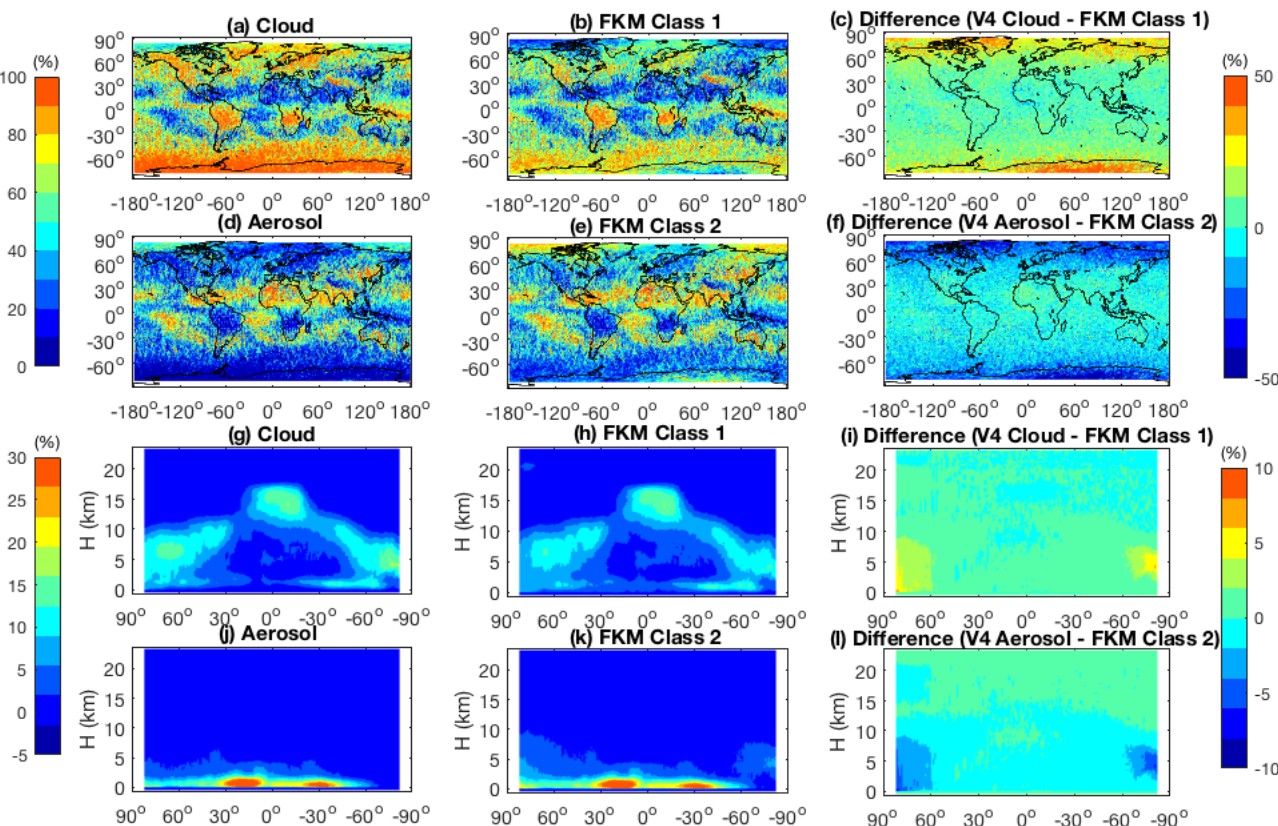

Figure 7: distributions of feature type occurrence frequencies during January 2008. Panels in the left column show V4 COCA results; panels in the center column show CAD$_{FKM}$ 2-class results; and the panels in the right column show the percentages of

differences between the left and center columns. The top two rows show maps of occurrence frequencies as a function of

latitude and longitude for clouds (panels a–c) and aerosols (panels d–f). The bottom two rows show the zonal mean occurrence frequencies of clouds (panels g–i) and aerosols (panels j–l).

Above, we qualitatively show the operational classification algorithm agrees well with FKM algorithm. To quantify the degree to which the different methods agree with each other, we construct confusion matrices, which use the January 2008 5-km merged layer data between 60°S and 60°N to calculate the concurrent frequency of cloud and aerosol identifications made by the COCA and $CAD_{FKM}$ algorithms. We summarize the occurrence frequency statistics in Table 2. From the table we find that for our test month COCA V3 agrees with COCA V4 CAD for 96.6 % of the cases. The agreements are around 90 % for the entire globe including regions beyond 60° (not shown here). The FKM 2-class and 3-class results agree with both V3 and V4 for more than 93 % of the cases. The FKM results agree slightly better with V3 than with V4. All algorithms and versions agree on cloud coverage of around 58 % to 66 % of the globe. These values are well within typical cloud climatology estimates of 50 % to 70 % (Stubenrauch et al. 2013). Compared to the 2-class $CAD_{FKM}$, results from the 3-class $CAD_{FKM}$ agree somewhat better with the classifications from both the V3 and V4 CAD algorithms. Consistent with previous results in this paper, the 3-class $CAD_{FKM}$ appears better able to separate clouds and aerosols than the 2-class $CAD_{FKM}$. Figure 4 provides an additional example. For those water clouds beneath dense smoke, the 3-class $CAD_{FKM}$ scores are substantially higher than both the 2-class $CAD_{FKM}$ scores and the operation V3 CAD scores, indicating that the 3-class $CAD_{FKM}$ algorithm correctly identifies these features with much higher classification confidence. While the discrepancies between the two techniques are pleasingly small, their root causes are still of some interest. For example, we note that the FKM algorithm shows a slight bias toward aerosols relative to the V4 COCA (a 2.4 % bias for the 3-class FKM versus a 1.5 % bias for the 2-class FKM). At present, we speculate that the bulk of these differences can be traced to the dichotomy between supervised (COCA) and unsupervised (FKM) learning techniques. Given scope and quality of the data currently available for use by the COCA and FKM methods, the correct classification of layers occupying the twilight zone separating clouds and aerosols remains somewhat uncertain, and hence different learning strategies are likely to come to different conclusions, even when provided the same evidence. We also calculated the concurrent occurrence frequencies for only those features with CI values less than 0.75 (or 0.5). When the data are restricted to only relatively high confidence classifications, the FKM results agree with V3 and V4 for better than 96 % (or 97 %) of the samples tested.

Table 2: Statistical confusion matrix of a 1-month (Jan. 2008) CAD analysis that shows the agreement percentages (detected as clouds: C, aerosols: A, or total of clouds and aerosols: T for both algorithms) between different methods (V3: version 3, V4: version 4, FKM: fuzzy K-means).

| Agreement (%) | | V4 | | | FKM (2-classes) | | | FKM (3-classes) | | |
|---|---|---|---|---|---|---|---|---|---|---|
| | | C | A | T | C | A | T | C | A | T |
| V3 | C | 66.1 | 2.1 | | 63.8 | 4.5 | | 64.6 | 3.7 | |
| | A | 1.2 | 30.5 | | 1.1 | 30.6 | | 1.9 | 29.9 | |
| | T | | | 96.6 | | | 94.4 | | | 94.5 |
| V4 | C | | - | | 59.3 | 4.9 | | 60.6 | 3.6 | |
| | A | | | | 1.5 | 34.4 | | 2.4 | 33.4 | |
| | T | | | | | | 93.6 | | | 94.0 |
| FKM (2-classes) | C | | - | | | - | | 60.1 | 0.6 | |
| | A | | | | | | | 2.8 | 36.5 | |
| | T | | | | | | | | | 96.7 |

## 4.4 Special cases study

In this section we investigate several of the challenging classification cases that motivated the extensive changes made in COCA in the transition from V3 to V4 (Liu et al., 2019). Comparisons are done for those cases between different algorithms and different algorithm versions to see how well each algorithm or version compares to "the truth" (i.e., as obtained by expert judgments). In addition to the dense smoke over opaque water cloud case shown in Figure 4, the CAD$_{FKM}$ algorithm, like the operational CAD algorithm, can occasionally have difficulty correctly identifying high altitude smoke, dense dust, lofted dust, cirrus fringes, polar stratosphere clouds (PSC) and stratospheric volcanic ash (Figure 8-10). We briefly review each of these cases below.

a. Dust

Two different dust cases are selected for this study (Fig. 8). The first case examines nighttime measurements of a deep and sometimes extremely dense dust plume in the Taklamakan desert beginning at 20:15:32 UTC on 4 May 2008, as shown in Figs. 8 a-e. The second case investigates spatially diffuse Asian dust lofted high into the atmosphere while being transported toward the Arctic during a nighttime orbit segment beginning at 18:28:54 UCT on 1 March 2008, as shown in Figs. 8 f-j. CAD classifications are color-coded as follows: regions where no features were detected are shown in pure blue; V3 stratospheric features are shown in black; cirrus fringes are shown in pale blue; aerosol-like features are shown using an orange-to-yellow spectrum, with orange indicating higher confidence and yellow lower confidence; and cloud-like features are rendered in gray scale, with brighter and whiter hues indicating higher classification confidence. Dust layers in Taklamakan exhibit high 532 nm attenuated backscatter coefficients, high depolarization ratios (not shown), and attenuated backscatter color ratios close to

1 (also not shown). As seen between ~ 44° N and ~ 40° N, layers with this combination of layer optical properties are frequently misclassified as ice clouds in COCA V3 (Fig. 8b). However, in COCA V4 these same layers are much more likely to be correctly classified as aerosol (Fig. 8c).The 2-class and 3-class $CAD_{FKM}$ classifications both agree with COCA V4 for the lofted aerosols, but misclassify the densest portions of the dust plume as low confidence cloud. For the lofted Asian dust case shown in Figure 8 f-j, COCA V3 frequently misclassifies dust filaments as cloud, whereas COCA V4 correctly identifies the vast majority as dust. (Note too that many more layers are detected in V4 as a consequence of the changes made to the CALIOP 532 nm calibration algorithms (Kar et al., 2018; Getzewich et al., 2018; Liu et al., 2019).) The 2-class and 3-class $CAD_{FKM}$ classifications are essentially identical to those determined by COCA V4, but show higher confidence values for the aerosol layers.

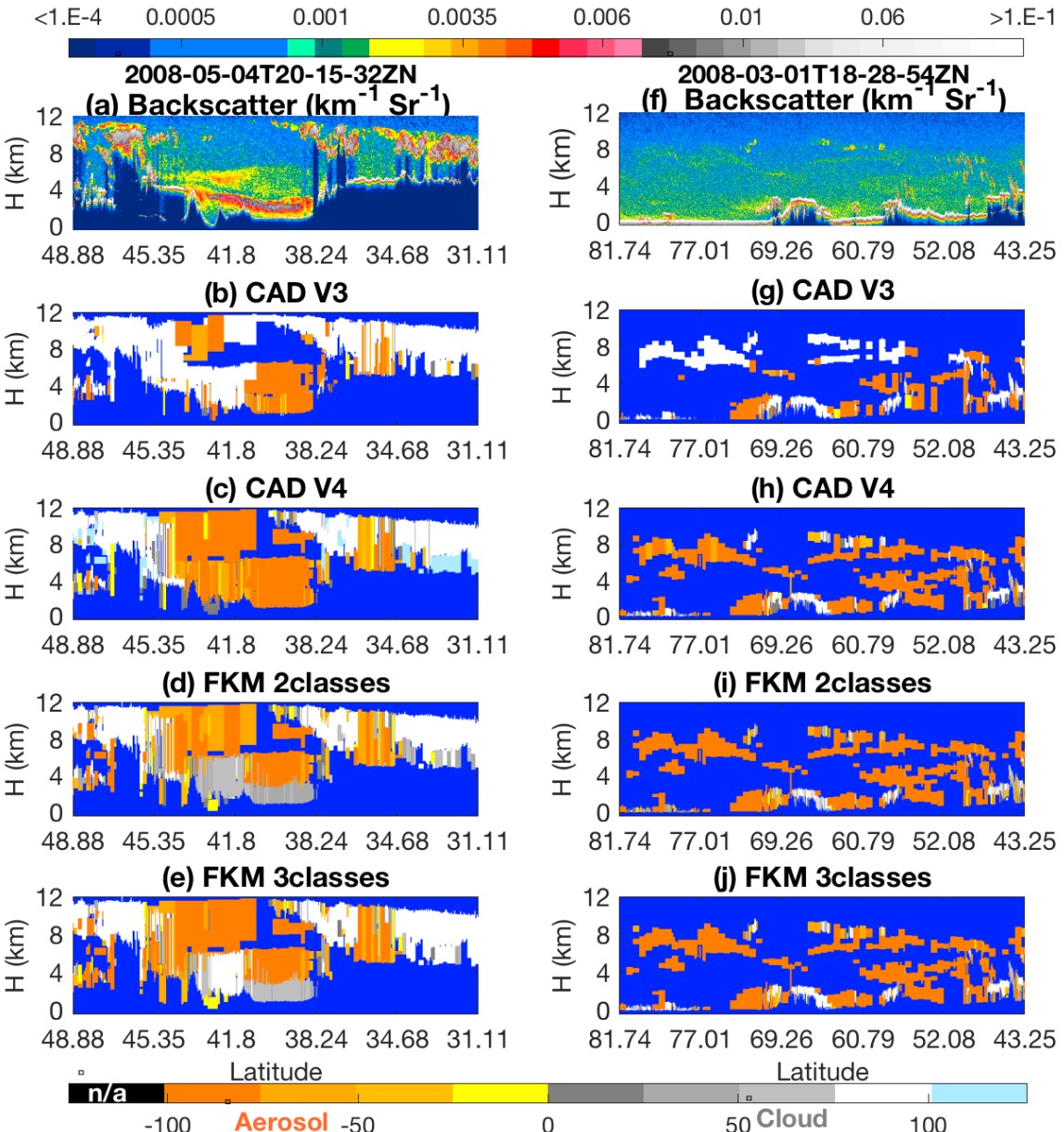

Figure 8: top row shows 532 nm attenuated backscatter coefficients for (a) dust in the Taklamakan basin on 4 May 2008 and (f) lofted Asian dust being transported into the Arctic on 1 March 2008. The rows below show the CAD results reported by four different algorithms; COCA V3 (panels b and g), COCA V4 (panels c and h), the 2-class $CAD_{FKM}$ (panels d and i), and the 3-class $CAD_{FKM}$ (panels e and j).

b. High altitude smoke

An unprecedented example of high-altitude smoke plumes was observed by CALIPSO during the "Black Saturday" fires that started 7 February 2009, quickly spread across the Australian state of Victoria, and eventually lofted well into the stratosphere (de Laat et al., 2012). Figure 9 shows extensive smoke layers at 10 km and higher on Monday 10 February between 20°S-40°S. In the V3 CALIOP data products, stratospheric layers (i.e., layers with base altitudes above the local tropopause) were not further classified as clouds or aerosols, but instead were designated as generic 'stratospheric features' (Liu et al., 2019). Consequently, COCA V3 misclassifies these smoke layers as clouds when their base altitude is below the tropopause and as stratospheric features when the base altitude is higher (Fig. 9b). On the other hand, the V4 CAD correctly identifies them as aerosols (Fig. 9c). In analyzing this scene we used two separate versions of the FKM algorithm. Our standard configuration used $z_{mid}$ as one of the classification attributions, while a second, trial configuration omitted $z_{mid}$. For the 2-classes FKM, both configurations successfully identified the high-altitude smoke as aerosol (Figs. 9d and 9e). But for the 3-classes FKM, including $z_{mid}$ as a classification attribute introduced uniform misclassification of the lofted smoke as cloud (Fig. 9f). However, when $z_{mid}$ is omitted the 3-class FKM correctly recognizes the smoke as aerosol (Fig. 9g). This is because including altitude information can introduce unwanted classification uncertainties when attempting to distinguish between high altitude clouds and aerosols, both of which are located at similar altitudes and have similar optical properties. Altitude is not a driving factor for classifications, and adds confusion in the memberships defined by the Mahalanobis distance (see Eqs. (7) and (8)) in these particular cases. More details are given in section 5.1. When high altitude depolarizing aerosols and ice clouds appear at the same time, either increasing the number of classes to four or omitting $z_{mid}$ as an input will resolve large fractions of the potential misclassifications from the FKM method.

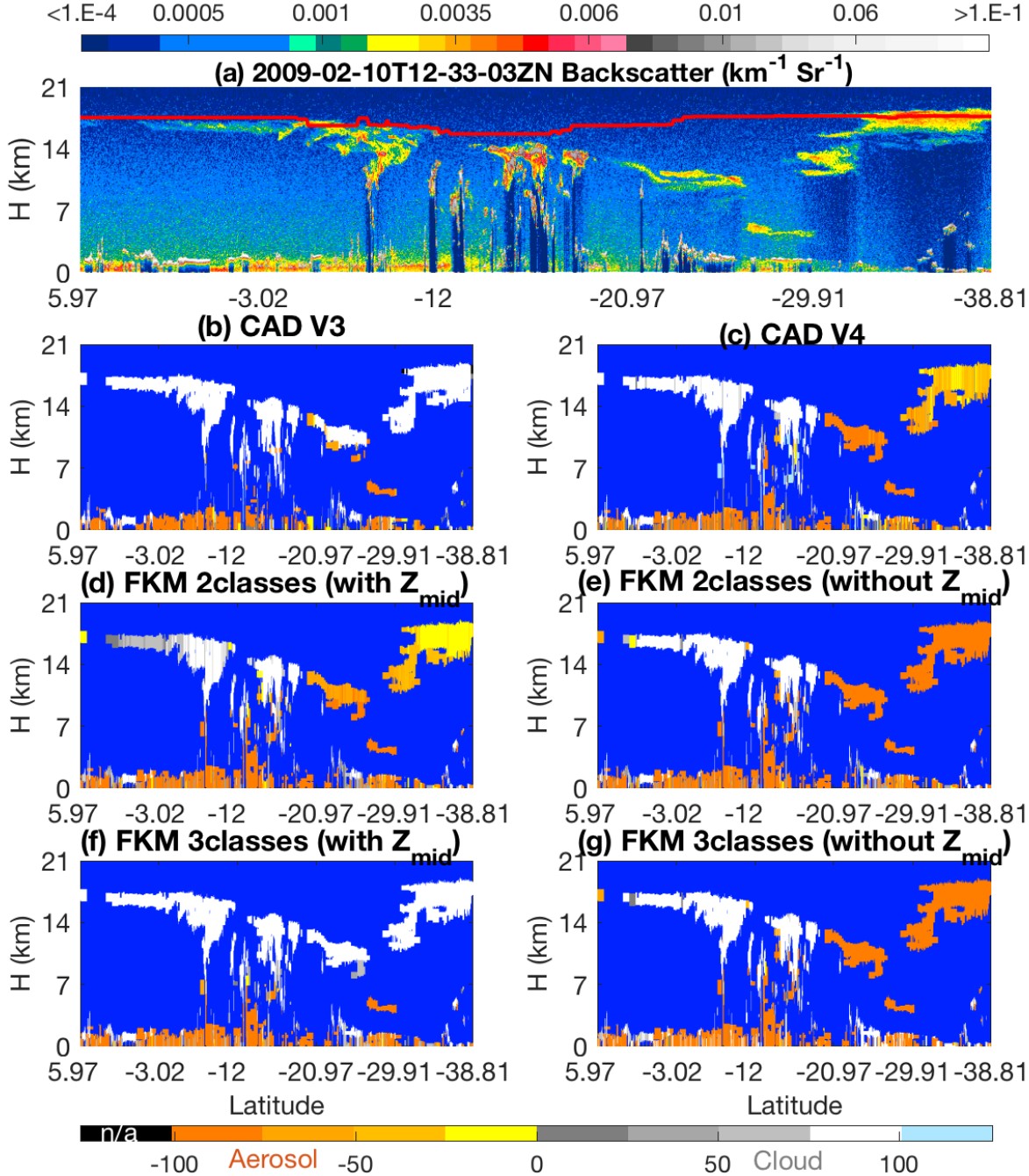

Figure 9: top row shows 532 nm attenuated backscatter coefficients for (a) measurements acquired on 10 February 2009 showing smoke injected into the upper troposphere and lower stratosphere by the Black Saturday fires in Australia. The solid line extending across (f) at altitudes between ~7 km and ~8.5 km shows the approximate tropopause altitude. The rows below

show the CAD results reported by six different algorithms; the V3 operational CAD (b), the V4 operational CAD (c), the 2-class $CAD_{FKM}$ with $z_{mid}$ (d) and without $z_{mid}$ (e), and the 3-class $CAD_{FKM}$ with $z_{mid}$ (e) and without $z_{mid}$ (j).

### c.  Volcanic ash

Figure 10 shows an example of ash from the Kasatochi volcano (52.2°N, 175.5°W), which erupted unexpectedly on 7–8 August

5   2008 in the central Aleutian Islands. Volcanic aerosols remained readily visible in the CALIOP images for over 3 months after the eruption (Prata, et al. 2017). On 5 October 2008, CALIOP observed the 'aerosol plume' near the tropopause at ~17:30:18 UCT. COCA V3 classified those layers with base altitudes above the tropopause as 'stratospheric features' (black regions in Fig. 10b), and misclassified a substantial portion of the lower, tropospheric layers as cloud. Those segments that were correctly classified as aerosol were frequently assigned low CAD scores. In contrast, COCA V4 and both versions of the $CAD_{FKM}$ with

10  $z_{mid}$ as inputs show greatly reduced cloud classifications, and the aerosols have high confidence CAD scores. Again, when altitude information is not included, the FKM algorithm produces a better separation of clouds and aerosols at high altitudes, for the same reasons as in the high altitude smoke case.

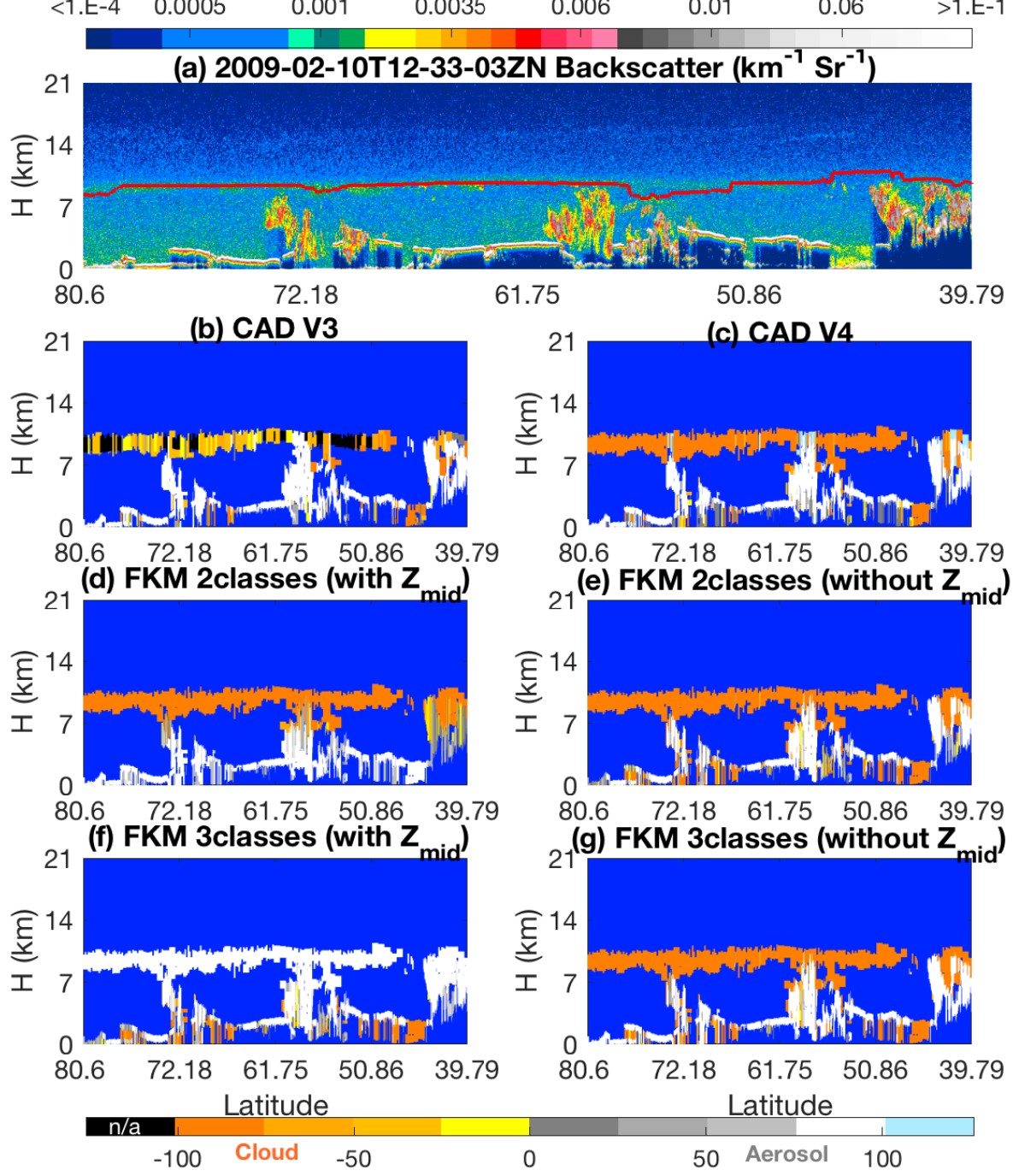

Figure 10: top row shows 532 nm attenuated backscatter coefficients for (a) measurements acquired on 5 October 2008 showing a layer of volcanic ash from the eruption of Kasatochi. The solid line extending across (f) at altitudes between ~7 km

and ~8.5 km shows the approximate tropopause altitude. The rows below show the CAD results reported by six different algorithms; the V3 operational CAD (b), the V4 operational CAD (c), the 2-class $CAD_{FKM}$ with $z_{mid}$ (d) and without $z_{mid}$ (e), and the 3-class $CAD_{FKM}$ with $z_{mid}$ (e) and without $z_{mid}$ (j).

## 5 Discussion

Section 5 compares FKM and COCA using statistical analyses and individual case studies. In this section, we explore the application of various metrics used to evaluate the quality of the FKM and COCA classifications. The questions we address are: (a) how much improvement can be made by adding additional measurements as classification inputs (Sec. 5.1); (b) how well are the classes separated (Sect. 5.2); (c) what are the essential measurements required for accurately discriminating between clouds and aerosols (Sect. 5.1 and Sect. 5.3); and (d) what effects do measurement uncertainties (noise) have on the
classifications (section 5.4)?

### 5.1 Key parameter analysis

Underlying any feature classification task is this essential question: which observations are most important for accurate feature identification? COCA results were substantially improved from V2 to V3 by adding two additional dimensions (latitude and volume depolarization ratio) to the cloud and aerosol PDFs. In general, higher dimension PDFs should improve the
classification accuracy so long as the additional dimensions provide some new useful information (i.e., they should be orthogonal, or at least semi-orthogonal, to the data already being used). It is therefore important to quantify how much improvement we can make by adding additional dimensions into the analysis. With the FKM method, it is relatively easy (though perhaps time-consuming) to add or remove one or multiple observational dimensions (i.e., inputs) and the reinitiate the training/learning algorithm. (This highly desirable flexibility is, unfortunately, wholly absent in the strictly supervised
learning regime incorporated into COCA.) If a dimension is added (or removed) and the new classifications are essentially identical to the old ones, the added (or removed) dimension does not provide significant information in the classification processes. On the other hand, if the CAD values are improved (or degraded) by adding or removing a dimension, this dimension actually contributes dispositive information in the determination of the classification, and hence is key to separating clouds from aerosols. By using the FKM method, we can readily determine which parameters are required (and, importantly, which
are non-essential) for the resulting classifications to meet predetermined accuracy specifications, either in general or for a particular class (e.g., dust). We can also quantify the improvement (or degradation) that occurs when specific parameters are either added or removed.

We demonstrate these capabilities using individual case study results. Figure 11 shows series of FKM classifications that omit individual dimensions from one half orbit of nighttime observations acquired 6 September 2008 (i.e., the same scene shown
previously in Fig. 4.) This scene was chosen as an example specifically because it contains so many challenging CAD cases

(e.g., PSCs, dense water clouds beneath smoke, and many high-altitude aerosols). Comparisons with COCA V4 and 2-class, 4-parameter $CAD_{FKM}$ results are quantified by the confusion matrices shown in Table 3. From both the figure and the table we find that the cloud-aerosol partitioning obtained when any one dimension is omitted is reasonably similar to the partitioning reported by COCA and by the 2-class, 4-parameter FKM algorithm. Both algorithms are most sensitive to the removal of $\chi'$ (75.0 % similarity for COCA and 77.4 % similarity for FKM), and least sensitive to the removal of $<\beta'_{532}>$ (89.8 % for COCA, 93.1 % for FKM). Note also from the figure we see that, for the low water clouds covered by a plume of heavy absorbing smoke, the 2-class 4-parameter FKM classifications have low $CAD_{FKM}$ values. When either $z_{mid}$ or $<\beta'_{532}>$ is removed from the classification parameters, the $CAD_{FKM}$ values actually improve. Without $\chi'$ or $\delta_v$, the $CAD_{FKM}$ values get worse, which indicates that color ratios and depolarization ratios may play a more important role in separating aerosols from low water clouds. In this example, the values of $\chi'$ and $<\beta'_{532}>$ measured in the water cloud can bias the resulting $CAD_{FKM}$ values due to the strong absorption at 532 nm within the overlying smoke layer. $<\beta'_{532}>$ for these water clouds decreases and gets closer to the backscatter magnitudes expected from classic aerosols (e.g., Figure 6a), while $\chi'$ increases far beyond values typical of classic aerosols. Moreover, when omitting $z_{mid}$, high altitude aerosols and ice clouds are more readily and correctly separated, as are low altitude aerosols and water clouds. $\chi'$ or $\delta_v$ are key in separating high altitude aerosols and clouds.

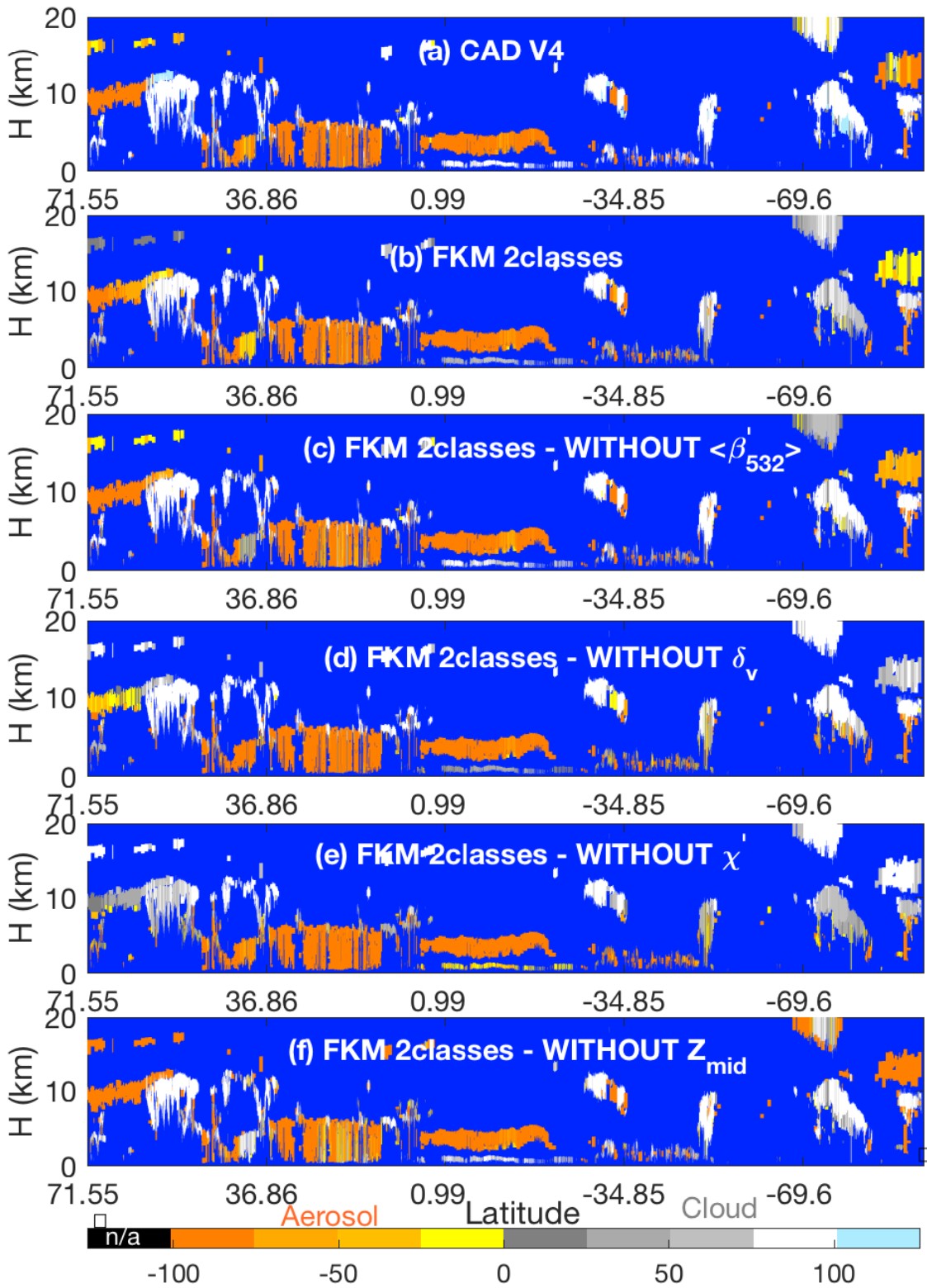

Figure 11: CAD scores calculated using various techniques for an orbit segment on 6 September 2008 beginning at 01:35:29 UTC. The upper two rows show results from (a) the V4 operational CAD algorithm and (b) the 2-class FKM algorithm using all four standard inputs. The remaining rows show 2-class CAD$_{FKM}$ results calculated when omitting one of the four standard inputs: (c) omits backscatter intensity ($<\beta'_{532}>$), (d) omits depolarization ratios, (e) omits color ratios, and (f) omits mid-layer height.

**Table 3: confusion matrices comparing COCA V4 and the CAD$_{FKM}$ results shown in Figure 11; abbreviations as follows: C = cloud; A = aerosol; and T = total.**

| (%) | | CAD$_{FKM}$ | | | CAD$_{FKM}$ (no $<\beta'_{532}>$) | | | CAD$_{FKM}$ (no $\delta_V$) | | | CAD$_{FKM}$ (no $\chi'$) | | | CAD$_{FKM}$ (no $Z_{mid}$) | | |
|---|---|---|---|---|---|---|---|---|---|---|---|---|---|---|---|---|
| | | C | A | T | C | A | T | C | A | T | C | A | T | C | A | T |
| **COCA V4** | C | 45.1 | 3.3 | | 45.6 | 2.8 | | 43.7 | 1.7 | | 40.1 | 11.4 | | 44.9 | 3.5 | |
| | A | 7.3 | 44.3 | | 7.4 | 44.2 | | 12.7 | 38.3 | | 13.6 | 34.9 | | 12.3 | 39.6 | |
| | T | | | 89.4 | | | 89.8 | | | 82.0 | | | 75.0 | | | 84.5 |
| **CAD$_{FKM}$** | C | | | | 56.9 | 2.6 | | 52.3 | 7.2 | | 49.6 | 10.0 | | 55.6 | 3.2 | |
| | A | | | | 4.3 | 36.1 | | 3.7 | 36.8 | | 12.6 | 27.9 | | 8.4 | 32.1 | |
| | T | | | | | | 93.1 | | | 89.1 | | | 77.4 | | | 87.8 |

In addition to the case study described above, we also analyzed a full month (January 2008) of CALIOP level 2 data acquired between 60°S and 60°N. To better focus on the troposphere, where the vast majority of detectable atmospheric layers occur, data from the polar regions were omitted in this test. We assessed the relative importance of various observational parameters by computing CAD$_{FKM}$ classifications using only a limited number of inputs (i.e., either 1, 2, or 3 of the CALIOP layer descriptors used in the standard 2-class CAD$_{FKM}$ classifications). These comparisons are summarized in Table 4. The center column of Table 4 shows the agreement frequencies of these classifications with COCA V4; the right column shows the agreement frequencies with the 2-class, 4-parameter FKM classifications.

Considering those comparisons where only one parameter is removed from the input data, it is clear that omitting $\chi'$ has by far the most deleterious effect. The classifications are relatively insensitive to omitting any of the other three parameters, though the comparisons are slightly worse when omitting $z_{mid}$ rather than $<\beta'_{532}>$ or $\delta_v$. The conclusions to be drawn from the single parameter classifications are similar to the 3-parameter case, though perhaps not as stark: using only $\chi'$ produced slightly better comparisons with both COCA V4 and the 2-class, 4-parameter CAD$_{FKM}$ results than any of the other parameters. Given this demonstrated sensitivity to $\chi'$, it is perhaps not surprising that of the 2-parameter classifications, the combination of $\chi'$ and $z_{mid}$ proves the most successful. The combination of $\chi'$ and $\delta_v$ also performed reasonably well relative to both COCA V4 and the

2-class, 4-parameter $CAD_{FKM}$. Unexpectedly, however, the combination of $\chi'$ and $<\beta'_{532}>$ performed very poorly relative to COCA V4, with only ~67 % of the classifications being identical.

In general, and as expected, the closest matches to the COCA V4 and the 2-class, 4-parameter $CAD_{FKM}$ classifications are achieved by the 3-parameter classifications, with the single parameter classifications showing the poorest correspondences, and the 2-parameter rankings falling somewhere in between the 3-parameter and 1-parameter results. However, the performance of the most successful 2-parameter case (the combination of $\chi'$ and $z_{mid}$) was largely identical to that of the most successful 3-parameter case (the combination of $<\beta'_{532}>$, $\chi'$ and $z_{mid}$). In fact, relative to COCA V4, the 2-parameter classifications were identical slightly more often (93.8 % of all cases) than the 3-parameter classifications (93.2 %). For the 2-class, 4-parameter FKM, the corresponding numbers rise to 95.5 % identity for the best performing 2-parameter classifications and 97.7 % identity for the best performing 3-parameter classifications. Both the COCA and FKM comparisons suggest that the addition of $<\beta'_{532}>$ adds little, if any, skill to the classification task but it contributes to the confidence of the classifications, as will be shown in Sect. 5.2.

**Table 4: Statistics of joint occurrence frequency during January 2008 from 60°S to 60°N between and the FKM classifications based on limited input parameter sets (i.e., 1, 2, or 3 CALIOP measurements, as listed in the left column), the COCA V4 classifications (center column), and the 2-class, 4-parameter (2-C, 4-P) $CAD_{FKM}$ classifications (right column).**

| Occurrence Frequency (%) | V4 CAD | 2-C, 4-P $CAD_{FKM}$ |
|---|---|---|
| $CAD_{FKM}$ ($<\beta'_{532}>$, $\delta_v$, $z_{mid}$, $\chi'$) | 93.61 | - |
| $CAD_{FKM}$ ($<\beta'_{532}>$, $z_{mid}$, $\chi'$) | 93.21 | 97.69 |
| $CAD_{FKM}$ ($\delta_v$, $z_{mid}$, $\chi'$) | 92.83 | 96.09 |
| $CAD_{FKM}$ ($<\beta'_{532}>$, $\delta_v$, $\chi'$) | 90.25 | 94.39 |
| $CAD_{FKM}$ ($<\beta'_{532}>$, $\delta_v$, $z_{mid}$) | 80.00 | 83.95 |
| $CAD_{FKM}$ ($\chi'$, $z_{mid}$) | 93.83 | 95.51 |
| $CAD_{FKM}$ ($\delta_v$, $\chi'$) | 90.96 | 93.77 |
| $CAD_{FKM}$ ($<\beta'_{532}>$, $\delta_v$) | 83.45 | 87.07 |
| $CAD_{FKM}$ ($<\beta'_{532}>$, $z_{mid}$) | 77.13 | 80.83 |
| $CAD_{FKM}$ ($\delta_v$, $z_{mid}$) | 75.66 | 79.42 |
| $CAD_{FKM}$ ($<\beta'_{532}>$, $\chi'$) | 66.89 | 70.11 |
| $CAD_{FKM}$ ($\chi'$) | 66.60 | 64.80 |
| $CAD_{FKM}$ ($\delta_v$) | 63.87 | 62.32 |
| $CAD_{FKM}$ ($z_{mid}$) | 63.77 | 62.06 |
| $CAD_{FKM}$ ($<\beta'_{532}>$) | 61.75 | 60.09 |

## 5.2 Fuzzy linear discriminant analysis

Linear discriminant analysis (Fisher, 1936) is usually performed to investigate differences among multivariate classes, to validate the classification quality, and to determine which attributes most efficiently contribute to the classifications. Here we introduce Wilks' lambda, which is the ratio of within-class variance (to evaluate the dispersion within class) and between-class variance (to examine the differences between the classes). Considering a data matrix X (n × p matrix, elements $x_{il}$, i, data number =1,..,n; l, data dimension number = 1,..p), the FKM classification returns a membership matrix M (n × k matrix, elements $m_{ij}$, i, data number= 1,..,n; j, class number = 1,..,k) and centroid matrix C (k × p matrix, elements $c_{jl}$, j, class number = 1,…,k; l, data dimension number = 1,…,p) where n is the number of data samples, p is the number of attributes/dimensions, and k is the number of classes. The sums of squares and products (SSP) within-classes covariance matrix $W_{lm}$ (p × p matrix, l/m, data dimension number = 1,…,p), also called the within-classes fuzzy scatter matrix (Bezdek, 1981), is given as

$$W_{lm} = \sum_{j=1}^{k}\sum_{i=1}^{n} m_{ij}^{\phi}\left(x_{il}-c_{jl}\right)\left(x_{im}-c_{jm}\right), \forall\left(l,m\right), l, m = 1,2...p \ . \tag{17}$$

The SSP between-classes covariance matrix $B_{lm}$ (*p × p* matrix, *l/m, data dimension number* = 1,…,*p*) are given as

$$B_{lm} = \sum_{j=1}^{k}\left(\sum_{i=1}^{n} m_{ij}^{\phi}\right)\left(c_{jl}-x_{l}\right)\left(c_{jm}-x_{m}\right), \forall\left(l,m\right), l, m = 1,2...p \ . \tag{18}$$

The ratio of within-classes to the total SSP matrix is known as Wilks' lambda (Eq. 19, Wilks, 1932). Wilks' lambda for multi-dimensional observations is the determinant of the p × p matrix, which represents the geometric volume of this object in *p* dimensions, written as

$$\Lambda = \frac{\det\left(W\right)}{\det\left(W+B\right)} \tag{19}$$

(Oh et al. 2005). Here we use Wilks' lambda ($\Lambda$) as a measure of the difference between classes. The value $\Lambda$ varies from 0 to 1, where 0 suggests that classes differ (within-classes SSP is smaller compared to between-classes SSP), and 1 suggests that all classes are the same. The magnitude of Wilks' $\Lambda$ indicates how distinct and well-separated the classes are. Smaller values of Wilks' $\Lambda$ indicate more distinct class separation with minimal between-class overlap thus the classification are more trustworthy and have higher confidence. Wilks' $\Lambda$ thus provides an additional metric to assess classification algorithm performance, augmenting the classification accuracy indicators shown in Sect. 5.1.

For the January 2008 data, Wilks' $\Lambda$ for different observational dimensions are calculated and summarized in Table 5. For 4-dimensional ($p$=4) observations, Wilks' $\Lambda$ could be as small as 0.21 for 2-class FKM and even smaller (0.05) for 3-class FKM. This means that the classes generated by the FKM method are well separated, with clusters quite different from each other, and that the classes in 3-class FKM are much better separated (less overlap in the multi-dimensional observations) than the

classes in 2-class FKM. For FKM 2-class, the value of Wilks' $\Lambda$ is largest for $z_{mid}$, indicating that, relative to the other individual parameters, clustering using $z_{mid}$ is less efficient at generating well-separated classes. The large value of Wilks' $\Lambda$ occurs because clouds have two distinct altitude centers, one for low water clouds and the other for high ice clouds. (Mid-level clouds occur too infrequently to form a third dominant altitude center.) The center altitude of water clouds is comparable to that of boundary layer aerosols, and thus it is very difficult to separate these two classes using $z_{mid}$ alone. The distinct altitude

centers of ice and water clouds induce large within-classes SSP and hence large values of Wilks' $\Lambda$. For single parameter clustering, Wilks' $\Lambda$ from $<\beta'_{532}>$ is the smallest, followed by the values for $\delta_v$ and $\chi'$. The value of Wilks' $\Lambda$ from any combination of observational dimensions lies between the maximum and minimum values for the single parameter clustering. Wilks' $\Lambda$ values for 3-class FKM are much smaller compared to the 2-class FKM values because $z_{mid}$ can have an independent center for each class. For 3-class FKM analyses, the largest single parameter values of Wilks' $\Lambda$ are produced by $\delta_v$, followed

by $z_{mid}$ and $\chi'$ . As with 2-class FKM, yields the smallest value.

**Table 5: Wilks' lambda ($\Lambda$) for 2-class (center column) and 3-class (right column) FKM classifications using different observational dimensions (left column).**

| Input parameters | $\Lambda$, 2 classes | $\Lambda$, 3 classes |
|---|---|---|
| $<\beta'_{532}>$, $\delta_v$, $\chi'$, $z_{mid}$ | 0.21 | 0.048 |
| $\delta_v$, $\chi'$, $z_{mid}$ | 0.20 | 0.060 |
| $<\beta'_{532}>$, $\chi'$, $z_{mid}$ | 0.20 | 0.060 |
| $<\beta'_{532}>$, $\delta_v$, $z_{mid}$ | 0.17 | 0.035 |
| $<\beta'_{532}>$, $\delta_v$, $\chi'$ | 0.14 | 0.030 |
| $<\beta'_{532}>$, $\delta_v$ | 0.12 | 0.025 |
| $<\beta'_{532}>$, $\chi'$ | 0.14 | 0.039 |
| $<\beta'_{532}>$, $z_{mid}$ | 0.14 | 0.052 |
| $\delta_v$, $\chi'$ | 0.13 | 0.043 |
| $\delta_v$, $z_{mid}$ | 0.20 | 0.056 |
| $\chi'$, $z_{mid}$ | 0.23 | 0.077 |
| $<\beta'_{532}>$ | 0.08 | 0.030 |
| $\delta_v$ | 0.16 | 0.136 |
| $\chi'$ | 0.18 | 0.053 |
| $z_{mid}$ | 0.28 | 0.121 |

## 5.3 Principal Component Analysis

In this section we apply principal component analysis (PCA; Wold et al. 1987) to the FKM classification results to determine which of the input parameters account for the greatest variability in the outputs. These functions, or canonical variants, are therefore calculated from the eigenvalues and eigenvectors of matrix $W_f / B_f$ (the ratio of within-class variance and between-class variance). The first function (PCA-1) maximizes the differences between the classes and represents the dominant contribution to the classifications. Successive functions (PCA-2) will be orthogonal to, or independent of, the other functions and hence their contributions to the discrimination between classes will not overlap. We also project the inputs variable vectors along the principal component axes. Using this method helps to better understand how independent the input parameters are and how they individually contribute to the classifications.

The scatter plots of PCA-1 and PCA-2 for FKM 2 classes and 3 classes are shown in Figure 12. The projection of vector lengths on PCA-1 and PCA-2 of different measurements (i.e., $<\beta'_{532}>$, $\delta_v$, and $\chi'$, and $z_{mid}$) indicate how much each individual dimension contributes to the classifications. Longer projections mean stronger contributions. From the figure, we clearly see that water clouds, ice clouds and aerosols are quite different (i.e., their cluster centers are located in different positions). Different colors represent different classes, and darker colors indicate higher sample densities. Class centers, marked with red crosses, are located where the class sample density is highest, with higher densities shown by darker colors. We reorient PCA-2 to keep the C1-C2 line approximately diagonal, and thus better assess the relationship between PCA-1 and PCA-2. (In reality, the contribution of PCA-1 is always larger than PCA-2 while the diagonal line shows PCA-1 contribution is equal to PCA-2.) From both panels we see that class 1 (cloud) of 2-class FKM breaks into 2 classes (ice cloud and water cloud) when applying 3-class FKM. The denser samples (centers) of water cloud, ice cloud and aerosol are quite separate from each other, and the overlap zone has fewer samples. We can also see that $\chi'$ and $\delta_v$ contribute the most to PCA-1 (longer projections on the axis of PCA-1 in both subpanels) while and $z_{mid}$ contribute more to PCA-2. Hence, $\chi'$ and $\delta_v$ are the driving components for the cloud-aerosol separation. From figure 12b, we could also argue that $<\beta'_{532}>$ and $\delta_v$ are the driving factors in classifying water and ice clouds (projections of the vectors on C2-C3, namely the combined projection of PCA-1 and PCA-2, are longer), while $\chi'$ and $z_{mid}$ also contribute to the classification. $z_{mid}$, and, to a greater extent, $\chi'$ and $\delta_v$ are the driving factors that allow aerosols to be separated from ice clouds (projections of the vectors on C1-C2 are longer), whereas  and $\chi'$ are the driving factors that separate water clouds from aerosols (projections of the vectors on C1-C3 are longer). Comparing contributions of individual measurements to different classes, $z_{mid}$ is most useful in helping discriminate aerosols and ice clouds, while simultaneously being the least useful in separating aerosols and water clouds. $<\beta'_{532}>$ is the most useful parameter in distinguishing aerosols form water clouds and water clouds from ice clouds, and the least useful in differentiating between aerosols and ice clouds. $\delta_v$ is most useful in distinguishing between water clouds and ice clouds and between aerosols and ice clouds, and the least useful

in separating aerosols from water clouds. These observations agree very well with earlier findings in Fig. 6 and Tables 2, 3 and 4.

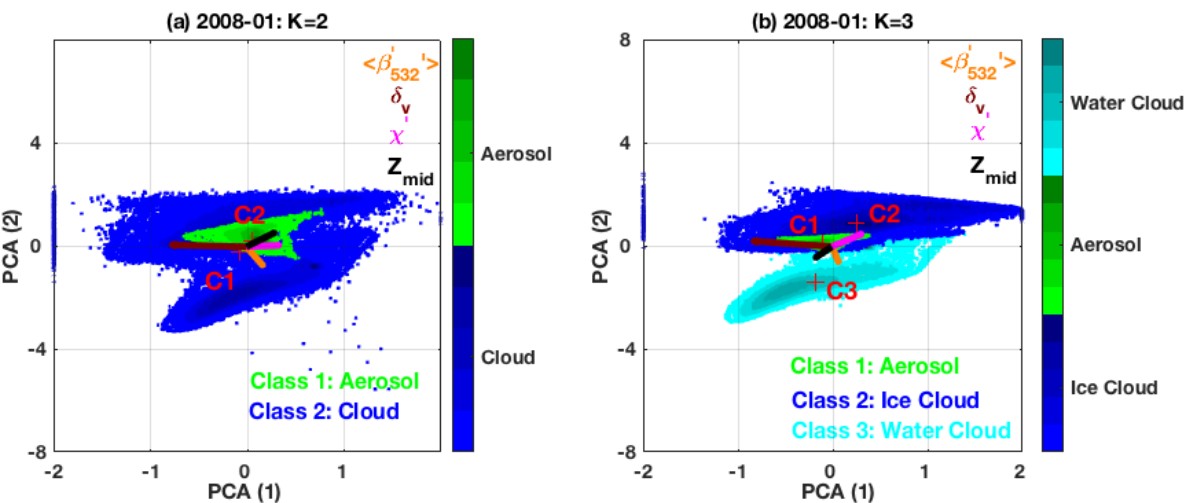

Figure 12: principle component analysis of the FKM classifications for the January 2008 test data. PCA results for the 2-class CAD$_{FKM}$ classifications are shown on the left (panel a), and the 3-class CAD$_{FKM}$ classifications are shown on the right (panel b). In both figures, the green points are projections of aerosol data onto the PCA axes with their center located at red crosses labeled C1. Similarly, the blue and cyan (left panel only) points are projections of cloud data onto the PCA axes. In panel a, the blue points represent all clouds, while in panel b the blue points represent ice clouds and the cyan points represent water clouds. Higher sample number condensations are in darker colors while lower condensations are in lighter colors. Also shown in both panels are color-coded vectors representing each of the classification variables: backscatter intensity ($<\beta'_{532}>$, in orange), depolarization ratio ($\delta_v$, in brown), color ratio ($\chi'$, in magenta), and altitude ($z_{mid}$, in olive). The projections of the variable vectors along the principal component axes indicate the degree to which each variable contributes to PCA1 and PCA2. Variable vectors that are parallel to either PCA1 or PCA2 contribute essential information to that component, while vectors that are perpendicular do not contribute at all.

## 5.4 Error propagation

By using PCA we can determine which parameters are most influential in arriving at different cluster memberships. Additionally, because all CALIOP measurements are contaminated to some degree by noise, we also want to see if/how noise in the individual parameters affects classification accuracy. These results can also guide us in understanding how the classification accuracy changes as the CALIOP laser energies deteriorate over the lifetime of the mission. In this section, we assess the impacts of instrument noise and measurement uncertainties on the FKM classifications. The observations from a nighttime granule acquired 6 September 2008 beginning at ~01:35:29 UTC are used to investigate how noise in the lidar

measurements affects the accuracy of the clustering results and what, if any, errors are introduced into the cloud and aerosol classifications for this particular case. To simulate the measurement uncertainties, two different methods are used. The first drew pseudo-random variables from Gaussian distributions having means equal to the various measured values and standard deviations between 10 % and 200 % of the means. As illustrated in Figure 13, using this method allows us to quantify the

effects of varying measurement errors on the FKM classification algorithm results. A sequence of Monte Carlo tests was constructed in which one of the four classification variables was randomly perturbed (i.e., drawn from the aforementioned Gaussian distributions) while the other three remained unchanged. For each of the four tests, 100 realizations of simulated input were created. To estimate the propagation of measurement uncertainties, we calculated the shifts in classification, confusion indexes (CI, see section 3.2), and the changes in cluster centers between new clusters with added noise and the

original clusters derived using unperturbed inputs. The shifts in cluster centers are the mean distances between the centers of the new clusters ($C_n$, obtained from perturbed dataset) and the old ones ($C_o$, obtained from error free dataset) for both clouds and aerosols, calculated using Eq. 20 (Omar et al., 2005). These distances are normalized by the standard deviation of the distributions ($C_{std}$) of individual record distances from unperturbed center as

$$\delta d_\varepsilon = \frac{|c_n - c_0|}{c_{std}} \tag{20}$$

where |x| represents the L1 norm of x. Figure 13 plots (a) the shifts in cluster centers for each class, (b) the fraction of correct classifications, and (c) the revised confusion indexes as a function of relative uncertainties ranging from 10 % to 200 %. From Figure 13a we see that shifts in cluster centers between perturbed and unperturbed data are very small when the uncertainties are small. The largest shift comes from color ratio perturbations and the smallest shift comes from backscatter perturbations. Perturbations on class-2 (aerosol) are more important compared to class-1 (cloud). Figures 13b and 13c show that when the

uncertainties in the measurements are small (i.e. less than 10 %), the errors in the classifications are also small (e.g., less than 2 % in Figure 13b) with less overlaps between classes (e.g., small values of CI from 0.3 to 0.305 seen in Figure 13c). When the uncertainties increase, the classification accuracies slightly decrease and the shifts in cluster center and CI slightly increase. The rates of change in the accuracy and confusion index are rapid at first (i.e., between relative uncertainties between 10 % and 100 %), but tend to stabilize for larger uncertainties. Large measurement uncertainties (i.e., 200 %) in color ratio can

introduce biases of 20 % in the classification results, with CI values less than 0.335. This suggests that uncertainties in the measurements can cause misclassification, but that most of the classifications (~80 %) are still robust. This is because cloud and aerosol properties are largely distinct and the misclassifications that do occur may come from features such as the few very thin clouds and dense aerosols in the transitional zone in Figure 6.

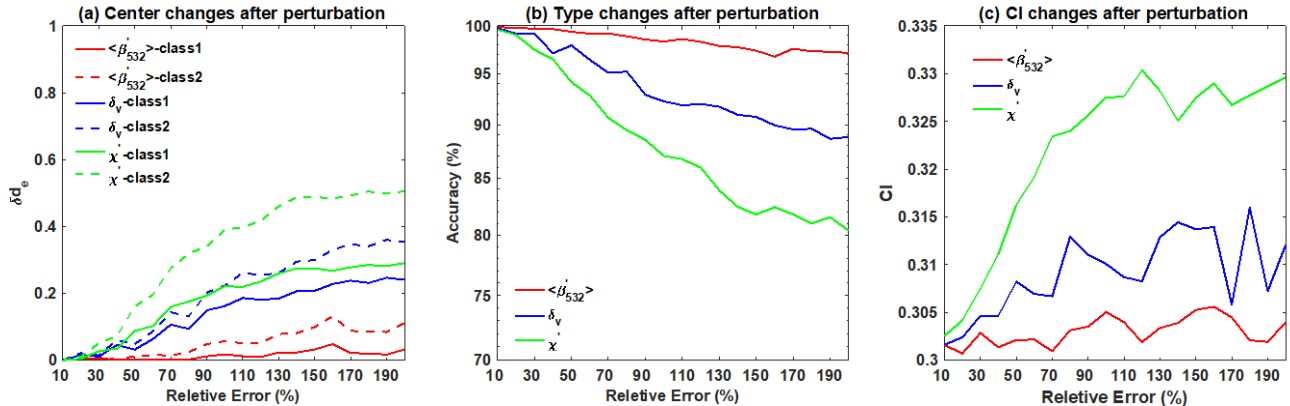

Figure 13: classification changes as a function of errors in the input parameters. The left panel (a) shows shifts in cluster centers for each class; the center panel (b) shows the relative accuracy of the FKM classifications; and the right panel (c) shows changes in the cluster confusion indexes. Panels b and c show perturbations in the classifications due to uncertainties in attenuated backscatter intensity ($<\beta'_{532}>$, in red), depolarization ratio ($\delta_v$, in blue), and color ratio ($\chi'$, in green).

Our first error propagation test used arbitrarily assigned relative uncertainties between 10 % and 200 % of the parameter mean values. In our second test we used the measured uncertainties reported in the CALIOP layer products to construct the Gaussian distributions from which pseudo-random variables were generated. By using this method, we can assess the actual impacts on the classifications due to noise in the CALIPSO measurements. To isolate the influence of the individual inputs, three test cases were constructed in which only one parameter was varied in each case. Figure 14 shows the results. Figure 14a shows the unperturbed results, while Figures 14b–14d show CAD$_{FKM}$ scores averaged over 100 perturbations of the test parameter. Figure 14b shows the results when the attenuated backscatter intensities are varied, Figure 14c shows the results when the depolarization ratios are varied, and Figure 14d shows the results when the color ratios are varied.

From Figure 14 we find that the averaged CAD$_{FKM}$ scores from the perturbed datasets do not differ markedly from the CAD$_{FKM}$ scores in the unperturbed dataset. In more than 88 % of the cases, clouds are still classified as clouds and aerosols are still classified as aerosols. When examining perturbations to backscatter intensity alone (Figure 14b), we find that the perturbed and unperturbed classification results are identical more than 98 % of the time. However, the CAD$_{FKM}$ differences arising from perturbations to depolarization ratio and color ratio (Figures 14c and 14d, respectively) can be much larger. This finding is consistent with results shown earlier in Figure 13.

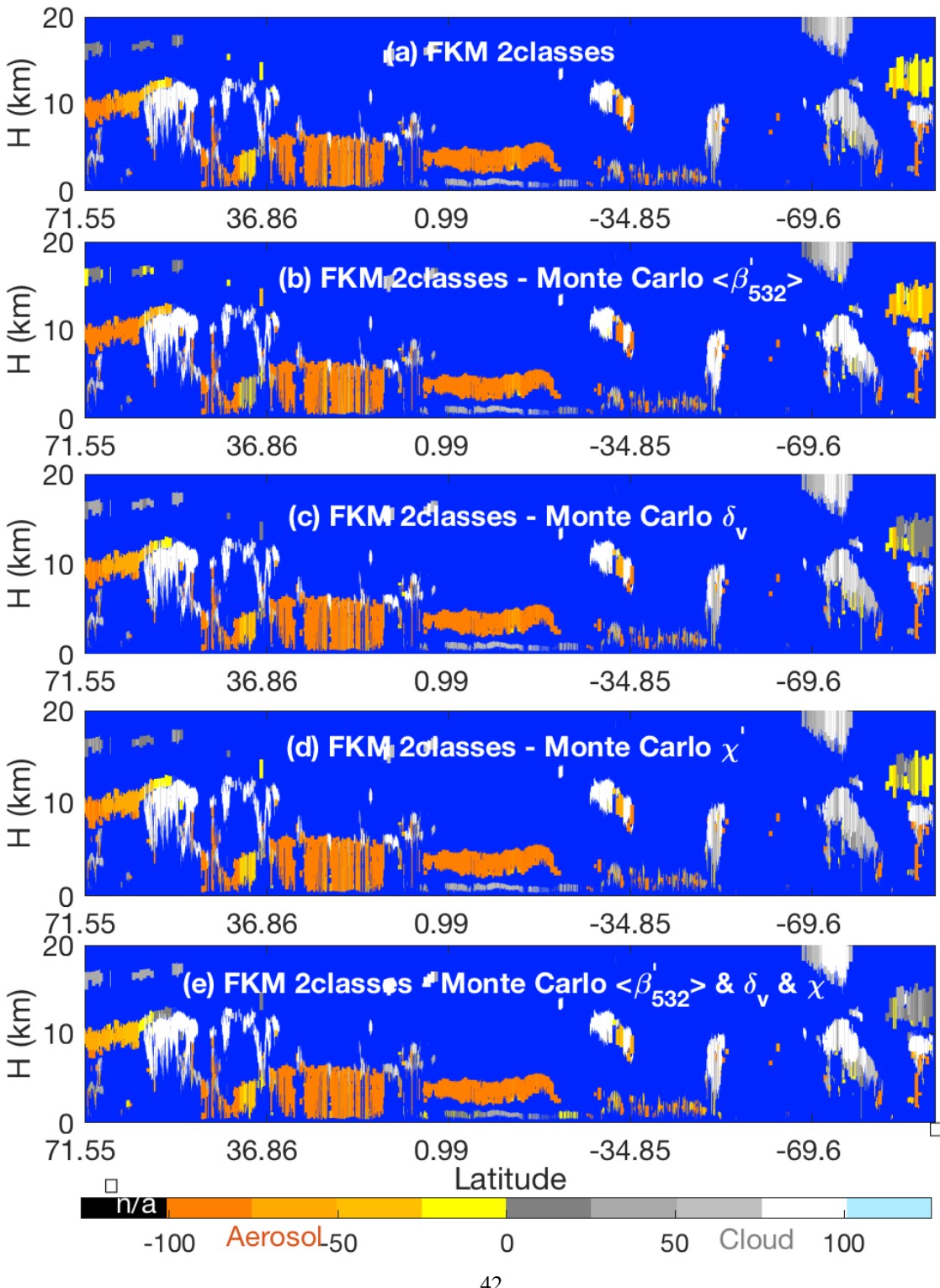

Figure 14: CAD scores for the same orbit shown in Fig. 4 (06 Sep. 2008, 01:35:29 GMT). The uppermost panel (a) shows 2-class FKM results derived using unperturbed measurements. Panels b, c, and d show, respectively, 2-class FKM results derived using perturbed measurements of attenuated backscatter intensity ($<\beta'_{532}>$), depolarization ratio ($\delta_v$), and color ratio ($\chi'$). Panel e shows 2-class FKM results derived when all three variables are perturbed independently.

Most often, the perturbed measurements only induce $CAD_{FKM}$ changes for features that were originally classified with low confidence and for those challenging features such as water clouds beneath smoke, high altitude aerosols, and PSCs, whose input parameters frequently lie in the transition zone between clouds and aerosols. Water clouds beneath thick smoke layers are an especially difficult case, as the uncertainties introduced by the absorption of smoke at 532 nm can significantly reduce the confidence of the water cloud classification. Looking at Figure 14, together with Figures 2, 11 and 12, we find that this is a reasonable and even expected result. From Figures 11 and 14 we know that the most effective measurements for separating water clouds and aerosols are color ratio and backscatter intensity. But relative to measurements of water clouds in otherwise clear skies, the color ratios for water clouds lying under absorbing smoke layers have large positive biases while the backscatter intensities have large negative biases, and these biases will produce low confidence CAD scores, both for the FKM method and the V4 operational method (Liu et al., 2019). A somewhat similar scenario can occur in the classification of high altitude aerosols, where high biases (i.e., measurement errors) in $\delta_v$ and $\chi'$ can lead to the misclassification of aerosols as ice clouds.

## 6 Conclusions

In this paper we use the fuzzy k-means (FKM) clustering algorithm to evaluate the classifications reported by the cloud-aerosol discrimination (CAD) algorithm used in the standard processing of the Cloud-Aerosol Lidar with Orthogonal Polarization (CALIOP) measurements. Being able to accurately separate clouds from aerosols is an essential task in the analysis of the elastic backscatter lidar measurements being continuously acquired by Cloud-Aerosol Lidar and Infrared Pathfinder Satellite Observations (CALIPSO) mission. When coupled to a well-validated CAD algorithm, the data products delivered by CALIOP can be used to reliably map the vertical distributions of clouds and aerosols on global and regional scales throughout the full 12 years of the CALIPSO mission.

The comparison between two different classification techniques helps us assess the performance of the operational CAD algorithm from the same scenes. The CALIOP operational CAD algorithm (COCA) is a supervised learning technique, in which classification decisions are tuned to match externally provided expert human judgements. FKM is an unsupervised learning scheme, which assigns class memberships based on similarities discovered in the inherent characteristics of the input data. While the two algorithms both rely on the same underlying lidar measurements, the underlying mathematical formulations are entirely different, as is the framework for expressing class membership values. These differences allow us to explore the classification uncertainties due to the algorithms. The flexibility of the FKM technique also allows us to investigate the relative

importance of various inputs in deriving the final classifications and to explore classification misclassification arising from current lidar measurement techniques. Establishing these performance metrics should enable the development of enhanced classification schemes for use with future space-based lidars.

The key finding of this study is that the feature classifications assigned by COCA are very closely replicated by the FKM method. Having a totally unsupervised learning algorithm "discover" the same patterns in the data that are reported by COCA strongly suggests that the COCA classifications represent genuine data-driven differences in the CALIOP observations, and are thus largely free from artifacts that might be imposed by human misinterpretations when constructing the CAD probability density functions (PDFs). The classifications obtained from our independently derived FKM analyses compare well with the classifications determined by COCA and reported in the CALIOP V4 data products. Using a one-month test set, the 2-class and 3-class FKM classifications agreed with the V3 and V4 operational data products over 93 % of the time, and the 3-class FKM results agreed with the COCA results in 94~95 % of all cases. This strong agreement between two independent methods provides convincing evidence that V4 operational CAD algorithm is delivering robust and accurate classifications.

Those instances where the two methods fail to agree (5~6 % of all cases) are typically highly ambiguous scenes in which the observables lie in the overlap regions between the peaks of the cloud and aerosol PDFs.  In particular, in scenes containing Taklamakan dust (or lofted Asian dust in general), high altitude smoke plumes, and/or volcanic ash, both the V4 operational CAD and the FKM algorithm struggle to make accurate classifications. The Taklamakan dust cases provide an instructive example that illustrates the classification conundrum. Over the Taklamakan, lofted dust layers and cirrus clouds occur in similar temperature regimes, and frequently have similar backscatter intensities ($<\beta'_{532}>$) and depolarization ratios ($\delta_v$). The most critical criterion for distinguishing clouds from aerosols is color ratio ($\chi'$), and the characteristic color ratios of dust and cirrus are reasonably distinct (~0.75 vs. ~1.01). However, the natural variability within each feature type is quite broad (e.g., ±0.25 for cirrus), and the measurements are very noisy, especially during daytime.

To characterize the CAD improvements made in the most recent CALIOP data release, we used the FKM method to explore the capabilities of both the V3 and V4 operational CAD algorithms. As expected, the V4 operational algorithm was more effective than the V3 version. The primary differences are found by examining the results obtained for specific feature classes. The FKM classifications agree well with both the V3 CAD results in most cirrus fringe and dense aerosol cases and agree well with V4 CAD results for lofted Asian dust, high altitude smoke, and volcanic ash. FKM classifications of stratospheric features and polar region features had the largest uncertainties. More studies are needed to better understand why these specific types of features are proving so resistant to confident classification, irrespective of the algorithmic approach applied.

Our investigation of error propagation in the FKM shows that while measurement uncertainties on the order of the CALIPSO measured noise will introduce biases into the cloud and aerosol classifications, more than 80 % of the classifications stay

unchanged. For the rest of classifications (which are low confidence clouds or aerosols), as the uncertainties increase, the classification confidence decreases, as indicated by higher confusion indexes, and the classification accuracies decrease as well. The dependence and the number of measurements can also impact the classification efficiency. Key parameter analysis shows that higher classification accuracies are achieved by increasing the number of independent observational parameters

used in the analyses. Two-class FKM classifications using only a single input yield the same results as the operational CAD in only ~60 % of all classifications, a rate only marginally better than would be expected from random choice. While using three parameters achieved an agreement between the FKM and the V4 operational CAD in the neighborhood of ~80 %, raising the agreement to ~95 % required four parameters. When only three inputs were used, removing color ratio from the FKM caused the largest classification disparities between the two methods.

Certain parameters are especially significant for the classification of particular feature types, and thus optimizing the number of successful classifications across all features requires the inclusion of all measurements that effectively contributed to any species-specific classification. Principal component analysis and key parameter analysis together show that the most important dimensions for distinguishing between clouds and aerosols are $\delta_v$ and $\chi'$; that $<\beta'_{532}>$ and $\delta_v$ are the driving factors in classifying water and ice clouds; and that altitude ($z_{mid}$), $\chi'$ and $\delta_v$ are the key inputs that allow aerosols to be separated from ice clouds,

while is the critical factor for separating aerosols and water clouds. Moreover, from fuzzy linear discriminant analysis we found the values of Wilks' lambda are close to 0, confirming that the FKM classification technique reliably separates clouds from aerosols.

The flexibility of FKM method offers opportunities to explore the effectiveness of future classification schemes that potentially incorporate measurements from multiple sensors, perhaps even from multiple satellites. While the input data used by our

implementation of the FKM technique is essentially synthetic to that required by the CALIOP V4 operational algorithm, the two decision-making frameworks are independently derived and rely on very different mathematics (i.e., probabilities vs. fuzzy logic). The very close similarity between the results produced by the two independent approaches argues strongly that the V4 operational classifications are essentially correct at the 94 % level.

*Data availability*: The analyses in this studied relied on the V4.10 CALIPSO level 2 vertical feature mask product (Vaughan et al., 2018b; NASA Langley Research Center Atmospheric Science Data Center; https://doi.org/10.5067/CALIOP/CALIPSO/LID_L2_VFM-Standard-V4-10; last access 22 May 2018) and the V4.10 CALIPSO level 2 5 km merged layer product (Vaughan et al., 2018b; NASA Langley Research Center Atmospheric Science

Data Center; https://doi.org/10.5067/CALIOP/CALIPSO/LID_L2_05kmMLay-Standard-V4-10; last access 22 May 2018). The CALIPSO level 2 data products are also available from the AERIS/ICARE Data and Services Center (http://www.icare.univ-lille1.fr, AERIS/ICARE; last access: 22 May 2018).

5 *Code availability*:  Source code for the fuzzy k-means algorithm used in this study was downloaded from the website of the University of Sydney:  https://sydney.edu.au/agriculture/pal/software/fuzme.shtml

*Competing interests*:  The authors declare that they have no conflicts of interest.  Author Charles Trepte is a co-guest-editor for the "CALIPSO version 4 algorithms and data products" special issue in Atmospheric Measurement Techniques but did not 10 participate in any aspects of the editorial review of this paper.

*Special issue statement*:  This article is part of the special issue "CALIPSO version 4 algorithms and data products". It is not affiliated with a conference.

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
