# Peer review of "Application of High-Dimensional Fuzzy K-means Cluster Analysis to CALIOP/CALIPSO Version 4.1 Cloud-Aerosol Discrimination"

_Atmospheric Measurement Techniques, 2018_

## Referee Comment (RC1) · Anonymous Referee #1 · 23 Jul 2018

Review of "Application of High-Dimensional Fuzzy K-means Cluster Analysis to CALIOP/CALIPSO Version 4.1 Cloud-Aerosol Discrimination" Shang Zheng, et al.

This manuscript describes a cluster analysis technique applied to CALIPSO data products as an alternative to the standard cloud-aerosol discrimination (CAD) algorithm(s). The stated objective of this activity is to validate the CAD algorithm and better understand what is important in the classification process.

I believe the elements of a good paper are here, but the logical flow needs some significant work. I need to really dig to understand why you did this in the first place, and what specifically you learned when you were done. The authors used a large number

of statistical and otherwise assessment techniques, but often gave little justification for the choices that were made during the assessment, and minimal description of the results. I think it would have been better to not describe every single analysis that was performed for this work, and instead focus on some of the most important and try for a better explanation of what you were doing, why you picked that technique, what you expected to find, and how the differences between expectations and results are significant to the CAD algorithms. The significance should be summarized both in the conclusions and the abstract. Before I start addressing specific portions of the manuscript, I have a few general questions and comments:

1. It seems to me that hierarchical clustering, in one of its many forms, would be useful to CALIPSO data. Presumably, clouds are very different from aerosols, but ice and water clouds, while different, are much more similar to each other than to aerosols. This is not the topic of the paper, of course, by if one is to spend time looking into clustering algorithms as an alternative for CAD, why wasn't this considered?

2. At no point do you discuss the possibility of the FKM acting as a potential replacement for the standard CAD algorithms. Why not? In the last paragraph, you mention that the "FKM method is much more time consuming than the operational algorithm." I don't understand this. Does this mean it is time consuming if one attempts to recreate the FKM for the entire dataset? I would think one would create the cluster centers with a subset of data and apply to the rest, which I couldn't imagine would be slower than the CAD approach. Other than speed, I see no discussion about why FKM would be less desirable than CAD. In some ways (less dependence on non scene information like latitude), I would think it would be preferable.

3. Much of your validation relies on a comparison of various versions of CAD and FKM. In some cases this is just for a few specific scenes. Once we make the leap of faith that those scenes are representative of all scenes, the fact remains that you don't know truth. So, is agreement between CAD and FKM the best metric when they could both be wrong? It seems the implicit assumption is that FKM could never reach CAD levels

of correctness, but when if FKM does correctly identify the scene but CAD does not? In that case, 'agreement' isn't appropriate as a means to validate the results. One way of addressing this problem is to apply both algorithms to synthetically generated scenes, where one knows the 'truth' and can verify its identification.

On to specific comments:

Abstract: As mentioned previously, the abstract gives minimal details about why you're undertaken this study, other than the rather vague "provide new insights" and validation of CAD. What are the new insights? Does it validate as expected?

Page 2, line 23: OK, so a difference between FKM and CAD is that the latter uses latitude as an input. I would think the arbitrary nature of the use of latitude is undesirable, so if FKM is successful than it's ability to perform without the use of latitude is very important and should be highlighted more.

Page 2: here you mention the difference between CAD V1 and CAD V4, but later on you validate against V3 and V4, and even mention V2 at some point. It seems overly complicated to compare against anything other than the latest version, but if you must you need to describe what is in each of the versions, the important differences, and why you need to validate against 3 & 4.

Page 4: OK, great, FKM doesn't use Latitude. Why not also skip altitude? Or try with and without? Altitude to me also seems like an arbitrary input that may bias your results, although perhaps somewhat more justifiable than latitude.

Page 5, last paragraph: You're using a random distribution of initial class memberships. How sensitive are you to that randomly selected distribution? Is there any difference in the results between one random seed and another? Also, is there any potential benefit in starting with class centers corresponding to preconceived notions of the class centers?

Figure 1, step 4: You have change in the norm of m as a metric to stop iteration (or

max iteration number). Could other metrics also be used, such as change in c, or d? I've seen iterative methods that use multiple means of assessing when no further improvement can be provided as a means to reduce the number of time the max iteration number boundary is used.

Page 7, paragraph 1: could the FKM be used to improve the PDF used in CAD for the 'arbitrary' classification inputs (latitude and altitude)?

Page 7, second paragraph: "...sample is used to determine the optimal number of classes and fuzzy exponent required for classification..." How is this done? This is an important detail to skip.

Figure 2: how were these PDF's defined? Also, caption needs to spell out HOI and ROI

Table 1: Based on the PDF's in Figure 2, it seems the filter criteria for AB is much tighter than what was selected for DR and CR. Why? Shouldn't they be similar?

Page 9, paragraph 1: "Mahalanobis distance can be used for correlated variables..." This seems to imply that you expect AB, DR and CR to be correlated, but can you make it clearer if they are or not? If they are (and I suspect this is the case), this has implications for the uncertainty analysis later.

Figure 3: (a) xaxis title should have caps (b) what is NCE? And more generally how do we know from these figures that the ideal # of classes is 3 or 4 and corresponding fuzzy exponents 1.4 or 1.6. It is not clear what I should be looking for in these plots.

Page 11, second paragraph: mentioned before, but again: is it essential to compare against both V3 and V4? It seems to be unnecessarily complicated.

Page 11, line 17 "...around 74S are misclassified..." confused by this statement since the lowest latitude on the figure is 71.44S

Page 13, line 15 "Note the value $(1-c1) \times 100$ for the 2-class FKM algorithm is equivalent

to the CADFKM score" isn't it equivalent to the absolute value of the CADFKM score?

Page 14, line 6 "It is evident..." While I agree they do look this way, there are much more rigorous comparison 'statistic' out there than eyeballing a figure.

Figure 7: it appears there is a big difference at high latitudes between FKM and CAD for clouds. Considering that CAD uses latitude, isn't this a very significant difference that should be highlighted and discussed further?

Table 2: What should I be looking for in the C and A columns and rows? This would be more meaningful if I knew what the actual expectations for percentage C and A should be.

Figures 4, 8, 9, 10, 11 would be easier to understand if the colorbar/label at the bottom indicated which side was 'aerosol like' and which was 'cloud like'.

Page 19, lines 12-13. This is an important point that must be emphasized! Use of altitude leads to mis-classification!

Page 21, line 6: this is the first time we hear that a reason for the use of V4 is improved calibration coefficients. Version differences for CAD should be described in more detail in an earlier section.

Figure 10 caption: spell out PSC, STS and NAT acronyms

Page 23, line 14: this is a confusing statement – you're using FKM to rebuild PDFs?

Page 23, line 20: under what circumstances does classification degrade with the addition of additional dimensions?

Figure 11: What is HH? I'm assuming altitude, but you use H elsewhere for that. It's probably best to spell out AB, DR, CR, HH in caption.

Page 25: I really don't understand why you are avoiding high latitude and altitude regions. This is where FKM could presumably help, and differences with CAD might

indicate problems with the latter's use of altitude and latitude as dimensions. Or, it might indicate that FKM without altitude and latitude still can't resolve well, in which case those dimensions are needed with CAD. This seems like a in important issue you're sidestepping.

Figure 12: How can you expect anybody to understand this figure? After a bit of effort, I think I understand what you're trying to show, but even if I am correct there's no way for me to differentiate the 15+ different colors. What should this figure look like in a perfect case? I think this figure and the corresponding text are an example of something that should be cut so more focus can be given to other sections.

Equation 16: It would be nice if you said a sentence about what this matrix should look like (ie square pxp matrix that is I in a perfect case)

Equation 18: What kind of norm is this?

Table 3. I'm having difficulty interpreting this. We want low Wilks' lambda for best classification, right? So lowest values are for backscatter alone for 2 class, and backscatter and depol for 3 class. So, does higher lambda when adding other parameters mean classification become worse? Or is it that wilks lambda can't be compared when the dimensionality is different? This is an example of an analysis that seems lacking in its description of what you are looking for, and what the results mean.

Section 4.3: why not just do PCA on the input parameters?

Figure 13: I'm confused by the distribution here. Are we to understand that aerosols are a class completely (or mostly) surrounded by other classes? Also, why these axis ranges, at least make PCA(2) from -4 to 4 so we can see what is going on.

Page 30, line 11: so, you're using one second of observations for the error propagation? I understand the computational limitations but this seems exceedingly limited.

Section 4.4 Aren't your dimensions correlated to some extent, such that you should expect uncertainties to be related (correlated) as well?

Equation 19: What kind of norm?

Figure 14: why not do this with actual CALIOP uncertainties, since these have been assessed? Also, why not use all three uncertainties simultaneously, like the real world?

Figure 15: why is the 2 class case the only one investigated this way?

Page 33, like 28: "While the two algorithms use largely identical inputs..." I heartily disagree with this statement. CAD uses altitude and latitude, which your analysis have shown to be important (even if you don't emphasize as much as I would like).

---

## Referee Comment (RC2) · Anonymous Referee #2 · 28 Sep 2018

The manuscript describes a methodology to discriminate between aerosol and cloud layers from CALIOP/CALIPSO lidar Level 2 data based on the high dimensional Fuzzy K-Means Cluster Analysis. The argument for sure is a good fit for the journal but some parts are not clear, probably suffering from hasty writing and need improvements before final publication. Moreover, other tests should be performed to improve scientific significance and clarity. I am however confident that the authors will brilliantly address all the issues I raised.

Major Comments:

The FKM clustering methodology is well described and totally makes sense. But, as

stated in the introduction, the FKM method is used to validate the result of V4 CAD algorithm and to better understand the classification, identifying the crucial parameters. It looks like that all the produced efforts have a very low return on investment. The V4 CAD is not validated vs. a reference dataset, i.e. using a synthetic lidar data where all the aerosol and cloud properties are well known and controlled, but with respect to another methodology that have comparable uncertainties. Moreover, It is completely missing an analysis on who is really using those data, i.e. climatologists, modelers. . ., and why it is critical to discriminate (defining a level of precision) between aerosols and clouds (and their subtypes). For example, how much is it the actual precision of the current operational V4 CAD algorithm in classifying the aerosol and cloud layers ? The final users are ok with this accuracy? Which benefits will be obtained reducing the misclassification? How the FKM will be used or implemented to reduce the V4 CAD misclassification?

In the manuscript is only marginally discussed why January 2008 measurement are a representative data sample. How the results are impacted changing the analyzed dataset ?

The number of classes is predefined (2 or 3) after analyzing Figure 3. However, in operational contexts, some data subsets might belong only to two classes. FKM still will fill with observation the class that should be empty. Is there a reason why the authors used the FKM cluster analysis instead of some self-selecting class methods, i.e. MeanShift clustering (Cheng, Yizong. "Mean shift, mode seeking, and clustering." IEEE transactions on pattern analysis and machine intelligence 17.8 (1995): 790-799) or classification algorithms as AdaBoost (Hu, Weiming, Wei Hu, and Steve Maybank. "AdaBoost-based algorithm for network intrusion detection." IEEE Transactions on Systems, Man, and Cybernetics, Part B (Cybernetics) 38.2 (2008): 577-583) ?

The random initialization of the centroids is a well-known problem as the initial centroid selection not only influences the efficiency of the algorithm, but also the number of relative iterations (and consequently the needed time machine). Some optimal centroid

selection techniques can be found in Nazeer, K.A. Sebastian, M.P Clustering biological data using enhanced k-means algorithm". In: Electronic Engineering and Computing Technology, Springer, 2010, pp. 433–442 (chapter 37)

Specific comments:

Line 27 Pag. 1 Please add also "geometrical properties"

Line 15 Pag. 5 How the random initialization influence the final result? I don't recall any section where this issue is discussed. Are the results consistent with the random initialization?

Line 16 Pag. 5 the authors mean Equations 2,3 and 4?

Figure 1: Third step it should be Eq. 6 and 7

Line 2 Pag. 7:: I am not sure that latitude is not useful to discriminate, as clouds at 16 km at polar latitudes may rise a flag, as cirrus clouds below 9km in the equatorial and tropical regions

Figure 3: labels are difficult to read. The picture in the middle shows "NCE" that is not previously defined.

Line 14 Pag 8: please rephrase "water clouds. For these water clouds".

Figure 4: it is very hard to see the zone of interest (smoke and cloud). Maybe reduce the vertical scale from 0 to 20 km?

Line 15 Pag 17 please read "We saw" instead of "We see"

Paragraphs 3.4 a, 3.4 b and 3.4 c. How the authors assume that the layer are pure dust, smoke and ash respectively ? Is there any other ancillary measurement that shows without any doubt the aerosol layer composition?

Section 4. Figure 13 is not very intuitive and it is difficult to get meaningful information from it . It might be interesting to replace it (or add) the Scree Plot and the loading

factors as barplot as showed in https://doi.org/10.1175/JTECH-D-15-0085.1.

Line 4 Pag. 34: Even if the FKM Cluster Analysis closely replicate the CAD V4 operational algorithm, it is not validate it (see main comment section)

Line 18 Pag. 35. FKM it is a time consuming algorithm because setting up random centroids can slow down the convergence process and in some cases can produce as result sub-optimal centroids virtual centroids (i.e. not corresponding to any observational measurement). See Main Comments section.

---

## Author Comment (AC1) · 7 Dec 2018

This manuscript describes a cluster analysis technique applied to CALIPSO data products as an alternative to the standard cloud-aerosol discrimination (CAD) algorithm(s). The stated objective of this activity is to validate the CAD algorithm and better understand what is important in the classification process.

I believe the elements of a good paper are here, but the logical flow needs some significant work. I need to really dig to understand why you did this in the first place, and what specifically you learned when you were done. The authors used a large number of statistical and otherwise assessment techniques, but often gave little justification for the choices that were made during the assessment, and minimal description of the results. I think it would have been better to not describe every single analysis that was performed for this work, and instead focus on some of the most important and try for a better explanation of what you were doing, why you picked that technique, what you expected to find, and how the differences between expectations and results are significant to the CAD algorithms. The significance should be summarized both in the conclusions and the abstract. Before I start addressing specific portions of the manuscript, I have a few general questions and comments:

Thanks for reviewing the manuscript and giving many valuable suggestions. We re-worked the logical flow of our paper, and we have clarified our working plan in the introduction. We better interlinked the sections and subsections to make the flow of the manuscript clear and easy for readers and reviewers to follow. We have also added more descriptions and explanations of the results in section 3 and 4, concentrated on and more clearly identified especially significant results, and better summarized them in the abstract and the conclusion. Please check details in the paper.

1. It seems to me that hierarchical clustering, in one of its many forms, would be useful to CALIPSO data. Presumably, clouds are very different from aerosols, but ice and water clouds, while different, are much more similar to each other than to aerosols. This is not the topic of the paper, of course, by if one is to spend time looking into clustering algorithms as an alternative for CAD, why wasn't this considered?

   It seems we've done a poor job of stating our objectives for this paper, as it was never meant to be an exercise in "looking into clustering algorithms as an alternative for CAD". Instead, as we say in the conclusions of the original draft, the purpose of this study is "to assess the performance of the cloud-aerosol discrimination (CAD) algorithm used in the standard processing". Having now read the referee's comments, it's quite apparent that we failed to clearly enunciate our primary goal.

   Unsupervised clustering schemes are frequently used to find patterns in data. The trick is to find patterns that have meaningful interpretations for human data users. Having a totally unsupervised learning algorithm "discover" the same patterns in the data that we report when using our CAD PDF scheme gives us some confidence that our CAD classifications represent genuine data-driven differences in the CALIOP observations, and are thus largely free from artifacts that might be imposed by human misinterpretations or conceits. Furthermore,

interpreting our comparisons of the results obtained by the two methods is straightforward: the higher the correspondence between the "discovered" classes and our predetermined classes, the higher our confidence in the CALIOP operational CAD classifications.

2. At no point do you discuss the possibility of the FKM acting as a potential replacement for the standard CAD algorithms. Why not?

Once again it appears that we have failed to convey our goals for this study. We are not investigating potential replacements for our CAD algorithm. Instead we are looking for methods to validate its performance and characterize its reliability. We will, of course, replace the standard CAD algorithms if/when some other algorithm(s) – be it FKM or anything else – can be shown to yield ***consistent and demonstrable improvement*** in distinguishing between clouds and aerosols ***on a global scale***. But that day has not yet arrived. The take-away message from this paper is (or least should be!) that the FKM classifications are essentially similar to the operational V4 results, and thus confirm that our operational CAD algorithms are performing quite well. In the testing we've conducted to date, we see no evidence of superior performance by the FKM algorithm.

In the last paragraph, you mention that the "FKM method is much more time consuming than the operational algorithm." I don't understand this. Does this mean it is time consuming if one attempts to recreate the FKM for the entire dataset? I would think one would create the cluster centers with a subset of data and apply to the rest, which I couldn't imagine would be slower than the CAD approach.

We were regrettably imprecise in our comments about algorithm speeds. What's time consuming is training the FKM on a suitably large subset of the measurements. (In this regard, FKM is similar to our standard algorithm: building the CAD PDFs is also a time- and compute-intensive activity.) However, once the FKM training has been completed, the partitioning of features within any scene into clouds and aerosols should occur on time scales commensurate with (or perhaps faster than) our current CAD algorithm.

Other than speed, I see no discussion about why FKM would be less desirable than CAD. In some ways (less dependence on non-scene information like latitude), I would think it would be preferable.

Conceptually, both FKM and the existing algorithm are potentially good candidates for the CAD task. But practically speaking, our existing algorithm is the hands down winner: it's already highly optimized for use with CALIOP measurements and tightly integrated into the CALIOP software architecture, and its performance has been extensively documented (e.g., see Liu et al., 2009 and Liu et al., 2018).

Regarding the use of "non-scene information like latitude", recall that the CALIOP CAD algorithm is fundamentally a probability-driven technique. And because (for example) the probability of observing lofted dust plumes at 60° S is significantly smaller than at 45° N, latitude can contribute relevant information in distinguishing cloud (e.g., cirrus) from aerosol (e.g., Asian dust). On the other hand, using attributes like latitude and altitude can introduce classification artifacts into ***unsupervised*** machine learning techniques like fuzzy k-means and Kohonen self-organizing maps. These techniques are likely to be much more successful if the inputs can be restricted to the intrinsic properties of the atmospheric layers being measured (e.g., particulate depolarization ratios, lidar ratios, Ångström exponents, etc.). Unfortunately, these quantities are not directly

measured by elastic backscatter lidars like CALIOP and thus cannot be used in the classification phase of the analysis. Instead we have to make due with proxies like total attenuated backscatter color ratio, which, while readily measured, is also a rather awkward combination of intrinsic and extrinsic layer properties. Having multi-wavelength high spectral resolution lidar measurements would remedy the situation…but, sadly, it's likely to be many years before such a capable system is flown in space.

3.  Much of your validation relies on a comparison of various versions of CAD and FKM. In some cases this is just for a few specific scenes. Once we make the leap of faith that those scenes are representative of all scenes, the fact remains that you don't know truth. So, is agreement between CAD and FKM the best metric when they could both be wrong? It seems the implicit assumption is that FKM could never reach CAD levels of correctness, but when if FKM does correctly identify the scene but CAD does not? In that case, 'agreement' isn't appropriate as a means to validate the results. One way of addressing this problem is to apply both algorithms to synthetically generated scenes, where one knows the 'truth' and can verify its identification.

    To say that "much of ɟour validation relies on a comparison of various versions of CAD and FKM" is perhaps being too kind. A more realistic assessment might be that **ALL** of our validation currently relies on these comparisons. To the best of our knowledge, there are no published, observation-based validation studies of the CALIOP CAD results. This paper is our first attempt to evaluate the CALIOP cloud-aerosol discrimination problem within a different mathematical decision-making framework. (We note, though, that since our manuscript first appeared in the AMT discussion forum, another algorithm-comparison study using state vector machines has been published; see Brakhasi et al., 2018 at https://doi.org/10.1016/j.jag.2018.07.017.)

    While the use of synthetic data as an evaluation tool is generally an excellent and highly effective strategy, in the case of discriminating clouds from aerosols it's also especially hard to implement in a useful way. For ~90% of the cases, cloud and aerosol properties are very well separated and reliable classifications can be made using a single wavelength elastic backscatter lidar (e.g., CATS; see https://cats.gsfc.nasa.gov/media/docs/CATS_QS_L2O_Layer_3.00.pdf). In these cases, unambiguous synthetic data can be used to weed out those algorithms that are obviously deficient. But the remaining cases fall into the cloud-aerosol overlap region (see Liu et al., 2009), where (to within the accuracy and precision of our measurements) aerosols and clouds layer can have essentially identical optical properties, and thus cannot be distinguished based only on the CALIOP measurements. Within the overlap region, the best an algorithm can hope to achieve is to avoid biases by reporting roughly equal correct and incorrect classification rates. (Ideally, all 'overlap region' classifications will be flagged as "low-to-no confidence"; i.e., assigned very low CAD scores). Unfortunately, given the available layer attributes, FKM cannot help resolve issues in the overlap region. To better resolve cloud and aerosol layers, additional measurements and/or information are needed.

    Even the most sophisticated human observers cannot always agree on the correct partitioning of those layers that occupy the overlap regions (e.g., see Koren et al., 2007; Tackett and Di Girolamo, 2009; Varnai and Marshak, 2011; Balmes and Fu, 2018). These especially difficult cases include separating thin cirrus from lofted Asian dusts, separating evaporating water cloud filaments from the surrounding aerosols in the marine boundary layer, and separating fresh volcanic ash from cirrus. Given the measurements available on the CALIPSO platform, the classification of these targets is always subject to some uncertainty. So by using synthetic data to compare algorithm outputs versus "truth" we could perhaps choose an algorithm that best confirms our own

prejudices; but whether that algorithm was delivering the correct classifications in the really hard cases would still be an open question.

On to specific comments:

Abstract: As mentioned previously, the abstract gives minimal details about why you're undertaken this study, other than the rather vague "provide new insights" and validation of CAD. What are the new insights? Does it validate as expected?

We added more detailed results as "insight" in the abstract. We also clarified that in addition to validating the CALIPSO operational CAD algorithm (COCA), the comparison work helps establish the boundaries of classification correctness regarding the individual classification algorithm. The comparison works in general as good as expected with more than 94% of classifications agreeing between different algorithms.

Page 2, line 23: OK, so a difference between FKM and CAD is that the latter uses latitude as an input. I would think the arbitrary nature of the use of latitude is undesirable, so if FKM is successful than it's ability to perform without the use of latitude is very important and should be highlighted more.

The latitude information is used in operational V4 CAD algorithm to create the probability density function (PDF) to discriminate the cloud from the aerosol. For every five-degrees of latitude and 1km altitude, a look-up table of 3-dimentional probability cloud and aerosol is built according to the joint distributions of backscatter, depolarization and color ratio observation from lidar. This is because for different zones and altitudes, the sources and dynamics of cloud and aerosol are different. Applying a single, global scale look-up table would make it extremely difficult to identify local features. Latitude itself is not an intrinsic optical characteristic that can be universally used to separate cloud and aerosol. But because the probability for a particular class to be present is location dependent, latitude helps to shape the PDFs to provide better classifications at a local scales.

The reason we don't use latitude as an input is because we use FKM method, which is a centroid classification method, and geographic information about clouds and aerosols doesn't provide good separation criteria (they both occur at all latitudes everywhere around the planet) and hence will confuse the FKM classification. When we add latitude as an additional input, the resulting Wilk's lambda rises to 0.5, indicating that the classifications are no longer reliable.

Page 2: here you mention the difference between CAD V1 and CAD V4, but later on you validate against V3 and V4, and even mention V2 at some point. It seems overly complicated to compare against anything other than the latest version, but if you must you need to describe what is in each of the versions, the important differences, and why you need to validate against 3 & 4.

We added explanations about the differences between version 3 & 4 CAD algorithms. We also explain briefly why we have chosen to compare FKM to both CAD versions.

Note too that our manuscript is a part of an AMT special issue on CALIPSO version 4 algorithms and data products, and thus is a companion to the paper by Liu et al. (2018) that describes the V3-to-V4 updates to the CALIOP CAD algorithm. Our comparisons of FKM to both V3 and V4 CAD are meant to provide additional insights into the improved performance of the V4 CAD.

Page 4: OK, great, FKM doesn't use Latitude. Why not also skip altitude? Or try with and without? Altitude to me also seems like an arbitrary input that may bias your results, although perhaps somewhat more justifiable than latitude.

The relative occurrence frequencies of aerosols and clouds are quite different, depending on altitude (and, to a lesser extent, latitude), and thus altitude is likely to be a highly relevant characteristic for supervised learning approaches. How relevant it is for unsupervised learning is somewhat less obvious. (One might argue that mid-layer temperature would be a much better choice, as in some cases (e.g., T > 0 °C) temperature can be a determining factor in distinguishing between ice clouds and water clouds.)

However, the reviewer is right that altitude is also not an intrinsic optical property that can be used to discriminate between clouds and aerosols. Instead it is an atmospheric dynamics property: aerosols and water clouds are most often found in the boundary layer, while ice clouds can reach as high as the tropopause.  So altitude can be crucial to discriminating between aerosol and ice clouds (though in some extreme cases their optical properties may be similar). For this reason we retain altitude in the FKM classifications. But, as the reviewer suggested, we have also experimented with omitting altitude for some particular cases in the paper (see section 4.4 in the revised manuscript). Throughout the paper we added more explanation saying why we choose these four parameters.

Page 5, last paragraph: You're using a random distribution of initial class memberships. How sensitive are you to that randomly selected distribution? Is there any difference in the results between one random seed and another? Also, is there any potential benefit in starting with class centers corresponding to preconceived notions of the class centers?

Actually, the code includes a loop for the selection of the random initiations so that the algorithm converges and gets the best fit.  We now clarify this point in the text and in the flowchart in Figure 1. We did not check the impacts of initiation distributions on classification results in the paper. To response to the reviewer's suggestion, we did a quick check on this and found the initial scattering does not impact the results much (please see the figures below). We used the uniform and normal random distributions for a 2-classes clustering and the results are almost the same; the difference for one orbit is 0.00000816%. While we could also use preconceived class centers, I don't think this will change the class membership value, but it might change the speed that the algorithm converges.

[Figure]

[Figure]

Figure 1, step 4: You have change in the norm of m as a metric to stop iteration (or max iteration number). Could other metrics also be used, such as change in c, or d? I've seen iterative methods that use multiple means of assessing when no further improvement can be provided as a means to reduce the number of time the max iteration number boundary is used.

We modified the flowchart and explanations about the last step (step 4) to determine our final membership due to our mistake in the previous manuscript. We actually used the objective function instead of membership which is not moving to determine when the iterations terminate. The code stops when algorithm converges and objective function change is smaller than a small threshold or the iteration count larger than a certain number. And we loop through multiple initiations and choose the smallest objective function minimum among all loop to get our final clusters. In real-world runs, the algorithm stops before reaching the max iteration number, which means the algorithm converged and objective function is stable. Note, the objective function is a function of membership, center and distance. That is why we say using only the center may slow down the speed. But I think technically we can use only the center, the distance or the membership. As the objective function takes joint account of all three, it would be our best choice.

Page 7, paragraph 1: could the FKM be used to improve the PDF used in CAD for the 'arbitrary' classification inputs (latitude and altitude)?

Perhaps…although to us it's not immediately clear how. In particular, the CAD altitude increments are chosen to partition the PDFs in a manner consistent with atmospheric dynamics. To a lesser extent, so too are the latitude increments. In fact, one of the primary reasons for increasing the latitude resolution from 10° in V3 to 5° in V4 was to achieve more reliable separation between ice clouds and dust in the northern hemisphere dust belt.

Page 7, second paragraph: "…sample is used to determine the optimal number of classes and fuzzy exponent required for classification. . ." How is this done? This is an important detail to skip.

We added more details about the selection of training data. We actually used one month of data (January 2008) to determine the optimal number of classes and fuzzy exponent required for classification.

Figure 2: how were these PDF's defined? Also, caption needs to spell out HOI and ROI

We added the definitions for PDFs in the text and the caption of Figure 2. We also add the definitions for HOI and ROI in the caption of Figure 2. The V4 CAD PDF is a five-dimension probability look-up table. Here the PDF is just the probability for one dimension, which is the occurrence frequency with sum normalized to unit.

Table 1: Based on the PDF's in Figure 2, it seems the filter criteria for AB is much tighter than what was selected for DR and CR. Why? Shouldn't they be similar?

Feature with AB larger than 0.2 are not majority of the cases and all of them are clouds. Aerosols with backscatter larger than 0.2 are relatively rare (e.g., only 3.5% of all tropospheric aerosol layers detected during 2012 at 5-km horizontal averaging had $<\beta'_{532}> > 0.2$ sr$^{-1}$ km$^{-1}$). Setting the value as 0.2 or 2 won't change the results much. When we set the values larger than 0.2 to 2, it won't change the classification results, but it will speed up the calculations and make it easy to converge. We added some more explanation about this in the paper.

Page 9, paragraph 1: "Mahalanobis distance can be used for correlated variables. . ." This seems to imply that you expect AB, DR and CR to be correlated, but can you make it clearer if they are or not? If they are (and I suspect this is the case), this has implications for the uncertainty analysis later.

As you suspect, the lidar observables are indeed correlated. If we consider each of the three as sums (or means) of the measured backscatter signal over some altitude range, then

$$\langle \beta'_{532} \rangle = \frac{1}{N} \sum_{n=1}^{N} \beta'_{532,\parallel}(z_n) + \beta'_{532,\perp}(z_n), \quad \delta_v = \frac{\sum_{n=1}^{N} \beta'_{532,\perp}(z_n)}{\sum_{n=1}^{N} \beta'_{532,\parallel}(z_n)}, \text{ and } \chi' = \frac{\langle \beta'_{1064} \rangle}{\langle \beta'_{532} \rangle} = \frac{\sum_{n=1}^{N} \beta'_{1064}(z_n)}{\sum_{n=1}^{N} \beta'_{532}(z_n)}.$$

(The subscripts $\parallel$ and $\perp$ represent contributions from the 532 nm parallel and perpendicular channels, respectively.) In particular, the signals measured in the 532 nm parallel channel contribute to all three quantities. We now provide these details in (the new) section 3.4.

Figure 3: (a) x-axis title should have caps (b) what is NCE? And more generally how do we know from these figures that the ideal # of classes is 3 or 4 and corresponding fuzzy exponents 1.4 or 1.6. It is not clear what I should be looking for in these plots.

X-axis is the class number (k), and NCE is MPE instead. We modified them in figure and its caption. We choose the value based on that fuzzy exponents is "best value of φ for that class is at the first maximum of objective function curves", which in subfigure c, for 3 classes (red color) the pick value corresponds to φ =1.4 and for 4 classes (light green) φ = 1.4 or 1.6. We choose the class number based on the minimum values of both FPI and MPE and also real class number in the atmosphere (cloud and aerosol).

Page 11, second paragraph: mentioned before, but again: is it essential to compare against both V3 and V4? It seems to be unnecessarily complicated.

Not absolutely essential, perhaps, but certainly highly desirable from our point of view. See our previous response on this topic for a bit more detail.

Page 11, line 17 ". . .around 74S are misclassified. . ." confused by this statement since the lowest latitude on the figure is 71.44S

It actually extended to 71.55N and about 80S but the figure only shows 69.6S. We modified "around 74S" to "beyond 69.6S"

Page 13, line 15 "Note the value (1-CI)x100 for the 2-class FKM algorithm is equivalent to the CADFKM score" isn't it equivalent to the absolute value of the CADFKM score?

We modified "the $CAD_{FKM}$ score" to "absolute value of the $CAD_{FKM}$ score".

Page 14, line 6 "It is evident. . ." While I agree they do look this way, there are much more rigorous comparison 'statistic' out there than eyeballing a figure.

We added statistics to support our "evident" claim.

Figure 7: it appears there is a big difference at high latitudes between FKM and CAD for clouds. Considering that CAD uses latitude, isn't this a very significant difference that should be highlighted and discussed further?

Further analysis of CAD in high latitude is out of the scope of work at the moment. In the future we can apply FKM for different altitude, latitude and season so as to improve the classification at local level. For different latitudes, the atmosphere dynamics are different, also the clouds and aerosol source and their intrinsic properties are quite different as well. To solve the problem, we can't simplify the problem by just adding/removing latitude with equal weighting. In this case sampled here, maybe the class number is not sufficient for the classification, so we may improve by adding classes number due to the large differences between stratospheric cloud and aerosol. This problem should be addressed in the future. Right now, the studies mainly discuss the comparisons of the classifications in the troposphere. We just want to highlight this information in the paper at the moment instead of trying to optimize FKM performance to resolve various problems. The paper already has a lot of information, and needs to stop somewhere so that we do not lose our focus (i.e., assessing the performance of the CALIOP operational CAD algorithm).

Table 2: What should I be looking for in the C and A columns and rows? This would be more meaningful if I knew what the actual expectations for percentage C and A should be.

The clouds global distribution is between 50%~70% according to Stubenrauch et al. (2013). For all algorithms or versions discussed in the paper, the cloud coverages are well within ranges that are considered acceptable. We added this reference in the paper text to support the comparison work here.

Figures 4, 8, 9, 10, 11 would be easier to understand if the colorbar/label at the bottom indicated which side was 'aerosol like' and which was 'cloud like'.

We add "cloud" and "aerosol" below the color bars to indicate cloud-like and aerosol-like colors. Since we still want to show the confidence of the classification, we added the following text just before

Figure 8: "CAD classifications are color-coded as follows: regions where no features were detected are shown in pure blue; fill values are shown in black; cirrus fringes are shown in pale blue; aerosol-like features are shown using an orange-to-yellow spectrum, with orange indicating higher confidence and yellow lower confidence; and cloud-like features are rendered in gray scale, with brighter and whiter hues indicating higher classification confidence."

Page 19, lines 12-13. This is an important point that must be emphasized! Use of altitude leads to mis-classification!

Yes, in some cases, using altitude will lead more misclassifications (e.g., in regimes having roughly equal numbers of samples of high altitude aerosol and ice clouds that also have highly similar optical properties) but for other cases (i.e., for a majority of cases, when separating ice clouds and low altitude aerosols) it will improve the classification. This is now discussed briefly in the last section. Altitude adds more confusion in separating water clouds from aerosols (e.g., when water cloud backscatter is small in optically thin water clouds), and in separating PSCs from stratospheric aerosols. Because these layers are at similar altitudes, altitude does not provide useful distinguishing information. But on the other hand, the altitude information can improve the classification between ice clouds and aerosols even if their depolarization and backscatter are similar. Ice clouds are high-altitude features, whereas aerosols are most often much lower. So it is hard to categorically say that the use of "altitude leads to misclassification". From Table 4, we also know that the combination of altitude and color ratio provides the best FKM classification compared to the V4 operational classification. It depends on the cases sampled and studied. While backscatter intensity can also produce confusion in the separation of thin clouds and dense aerosols, this occurs in much fewer cases. In majority of the cases, altitude helps improve the separation. We don't want to make an absolute conclusion just because of this case.

Page 21, line 6: this is the first time we hear that a reason for the use of V4 is improved calibration coefficients. Version differences for CAD should be described in more detail in an earlier section.

We added some brief explanations of the differences between V3 and V4 in the introduction. Detailed explanations of the differences between V3 and V4 are given in Liu et al. (2018), which we expect to be published part of this same AMT special issue. Our focus in this manuscript is on FKM and the comparison of the operational products with FKM products.

Figure 10 caption: spell out PSC, STS and NAT acronyms

We deleted the subsection discussing PSCs because the primary focus of our paper is on comparisons in troposphere.

Page 23, line 14: this is a confusing statement – you're using FKM to rebuild PDFs?

The referee is 100% correct: as originally written, this is indeed a confusing statement. To clarify, we have replaced the original text with the following revision.

Original
    With the FKM method, it is easy to add or remove one or multiple observational dimensions and re-cluster without re-building new PDFs.

Revised
With the FKM method, it is relatively easy (though perhaps time-consuming) to add or remove one or multiple observational dimensions (i.e., inputs dimensions) and the reinitiate the training/learning algorithm. (This highly desirable flexibility is, unfortunately, wholly absent in the strictly supervised learning regime incorporated into COCA.)

Page 23, line 20: under what circumstances does classification degrade with the addition of additional dimensions?

According to my understanding, it really depends on the case. This is also explained in the last section for the EOF analysis. If the dimension does not contribute information that helps segregate the data into distinct classes, it may instead degrade the accuracy of the classification. As an entirely speculative example, suppose we were to add UTC time-of-day as an additional input dimension. Since CALIPSO has a 16-day orbit repeat cycle, this sort of "information" may introduce subtle (or even not so subtle) artifacts into the derived clusters. To provide further clarification on this issue, we have added a pointer to additional discussion later in the manuscript (i.e., "details provided in Sect. 4.3").

Figure 11: What is HH? I'm assuming altitude, but you use H elsewhere for that. It's probably best to spell out AB, DR, CR, HH in caption.

We spelled out AB, DR, CR and H in the caption and change the "AB, DR, CR, and HH" in the title to "$<\beta'_{532}>$, $\delta_v$, $\chi'$, and $z_{mid}$" to keep all symbols in this paper same as those in the CAD paper published by Liu et al. 2018.

Page 25: I really don't understand why you are avoiding high latitude and altitude regions. This is where FKM could presumably help, and differences with CAD might indicate problems with the latter's use of altitude and latitude as dimensions. Or, it might indicate that FKM without altitude and latitude still can't resolve well, in which case those dimensions are needed with CAD. This seems like a in important issue you're sidestepping.

At high altitudes or latitudes, the intrinsic properties of clouds and aerosols are often much different from those in the other areas of the globe. Consequently, 2 or 3 classes may be not enough for the classification at a global scale. To improve the performance, we most likely need to apply FKM at local scales. And while that has not been done in this paper, it is worth further study in the future. This paper is not meant to be an exhaustive analysis, but should instead be seen as an initial step in an on-going CAD validation study. Future, more focused investigations of CAD performance in polar regions and the stratosphere are highly desirable next steps.

Figure 12: How can you expect anybody to understand this figure? After a bit of effort, I think I understand what you're trying to show, but even if I am correct there's no way for me to differentiate the 15+ different colors. What should this figure look like in a perfect case? I think this figure and the corresponding text are an example of something that should be cut so more focus can be given to other sections.

We deleted the figure. Instead, we used a table to summarize the numbers so as they are more easily understood by the reviewers and other readers. These numbers provide information all of the

information originally presented in the figure (e.g., how much improvement is made if we changed the inputs dimension from 3 parameters to 4 parameters, etc.).

Equation 16: It would be nice if you said a sentence about what this matrix should look like (i.e square pxp matrix that is I in a perfect case)

We added more the dimension information in the paper text i.e. "(k x p matrices)" in the text.

Equation 18: What kind of norm is this?

Equation 18 is not a norm, but instead shows the ratio of the determinant of W and the determinant of W + B. To clarify this, we have replaced the |W| notation with det(W). Determinates represent the volume of matrices in multiple dimensions while norms quantify distance in multiple dimensions.

Table 3. I'm having difficulty interpreting this. We want low Wilks' lambda for best classification, right? So lowest values are for backscatter alone for 2 class, and backscatter and depol for 3 class. So, does higher lambda when adding other parameters mean classification become worse? Or is it that wilks lambda can't be compared when the dimensionality is different? This is an example of an analysis that seems lacking in its description of what you are looking for, and what the results mean.

The best classification needs to consider many factors. The Wilks' lambda is only one indicator to help assess whether the clusters are sufficiently distinct or not. If clusters have no boundary between each other, although the classification accuracies are high, the classification results can't be trusted either. The classification results can be trusted when the accuracy is high as well as the Wilks' lambda is small, namely clusters are distinct to each other. I don't think that the dimension number itself will cause the increase of Wilk's lambda, it is how well a certain dimension bring in useful information for the distinction of the classification that matters. For example, the dimension of altitude adds more ambiguity to the classification confidence for the 2-classes classification because clouds are found at both high and low altitudes while aerosol are mostly found at low altitudes. For majority of the cases the backscatter intensity and depolarization ratio won't contribute ambiguity for the 2-classes classification and the backscatter intensity and color ratio won't contribute ambiguity for the 3-classes classification. We have to jointly looking at how well the classification does according to all the subsections in the sections. Generally speaking, smaller values indicate higher confidence in the results. But once the dimension brings in fuzzy factor (to make the boundary overlap a lot), although the accuracy increases, the classification Wilks' lambda will decreases. We added more explanation in the paper to clarify this.

Section 4.3: why not just do PCA on the input parameters?

We wanted to see what makes the classification distinct, so we have to use output instead of input to do PCA. We use Wilk's lambda to do PCA to see how well the classification results are. If we use inputs to do PCA, it may indicate how independent the inputs are which will answers the question on page 9 paragraph 1 from the reviewer. We reorganized the phrase to make it clear.

Figure 13: I'm confused by the distribution here. Are we to understand that aerosols are a class completely (or mostly) surrounded by other classes? Also, why these axis ranges, at least make PCA (2) from -4 to 4 so we can see what is going on.

It doesn't means that aerosol is completely surrounded by other classes. It just means aerosols are completely separated classes, as you can see the more condensed samples (darker colors, or centers in red crosses) of each class are distinct from each other. For example, thin clouds and dense aerosols are more similar to each other, so the aerosol cluster will have overlapped zone with clouds but the overlapped samples are less (lighter colors). We added more details about the colorbar in the paper to explain this. To let reviewer to clearly see the relationship between PCA2 and PCA1, we have reoriented the axis. This is because PCA1 contribution to the classification is more significant compared to PCA2 so that C1-C2 line is approximately in diagonal when PCA1 and PCA2 contribution is equal. We added more details in the paper.

Page 30, line 11: so, you're using one second of observations for the error propagation? I understand the computational limitations but this seems exceedingly limited.

Yes, the original text was not clear on that point. What we claimed was that "the observations from 6 September 2008 at ~01:35:29 UTC are used", and certainly that implies that we used only one second of data. What we used, however, was a full granule of data. Our revised sentence now says "the observations from *a nighttime granule acquired* 6 September 2008 *beginning* at ~01:35:29 UTC are used"

Section 4.4 Aren't your dimensions correlated to some extent, such that you should expect uncertainties to be related (correlated) as well?

Yes, the inputs are correlated, so the covariance terms in the uncertainties could either magnify or mask our current uncertainty estimates. The paper is not designed to check those.

Equation 19: What kind of norm?

Equation 19 is the L1 norm. We now say so explicitly in the revised manuscript.

Figure 14: why not do this with actual CALIOP uncertainties, since these have been assessed? Also, why not use all three uncertainties simultaneously, like the real world?

Figure 14 used the actual CALIOP uncertainties for each CALIOP observations. We also added, as the reviewer suggested, the results from the combination of the three uncertainties occurring simultaneously, as would be seen in real world.

Figure 15: why is the 2-class case the only one investigated this way?

Because the paper mainly discusses the CAD, which is a two class partitioning of the identified layers.

Page 33, like 28: "While the two algorithms use largely identical inputs…" I heartily disagree with this statement. CAD uses altitude and latitude, which your analysis has shown to be important (even if you don't emphasize as much as I would like).

The revised text now says "While the two algorithms both rely on the same underlying lidar measurements…", which is entirely accurate.

**References**

Balmes, K. A., and Fu, Q.: An Investigation of Optically Very Thin Ice Clouds from Ground-Based ARM Raman Lidars, Atmosphere, 9, 445; doi:10.3390/atmos9110445, 2018.

Brakhasi, F., Matkan, A., Hajeb, M., and Khoshelham, K.: Atmospheric scene classification using CALIPSO spaceborne lidar measurements in the Middle East and North Africa (MENA), and India, International Journal of Applied Earth Observation and Geoinformation, 73, 721-735, https://doi.org/10.1016/j.jag.2018.07.017, 2018.

Koren, I., Remer, L. A., Kaufman, Y. J., Rudich, Y., and Martins, J. V.: On the twilight zone between clouds and aerosols, Geophys. Res. Lett., 34, L08805, doi:10.1029/2007GL029253, 2007.

Liu, Z., Vaughan, M., Winker, D., Kittaka, C., Getzewich, B., Kuehn, R., Omar, A., Powell, K., Trepte, C., and Hostetler, C.: The CALIPSO lidar cloud and aerosol discrimination: Version 2 algorithm and initial assessment of performance, J. Atmos. Oceanic Technol., 26, 1198-1213, https://doi.org/10.1175/2009JTECHA1229.1, 2009.

Liu, Z., Kar, J., Zeng, S., Tackett, J., Vaughan, M., Avery, M., Pelon, J., Getzewich, B., Lee, K.-P., Magill, B., Omar, A., Lucker, P., Trepte, C., and Winker, D.: Discriminating Between Clouds and Aerosols in the CALIOP Version 4.1 Data Products, Atmos. Meas. Tech. Discuss., https://doi.org/10.5194/amt-2018-190, in review, 2018.

Tackett, J. L., and Di Girolamo, L.: Enhanced aerosol backscatter adjacent to tropical trade wind clouds revealed by satellite-based lidar, Geophys. Res. Lett., 36, L14804, https://doi.org/10.1029/2009GL039264, 2009.

Varnai, T. and Marshak, A.: Global CALIPSO Observations of Aerosol Changes Near Clouds, IEEE Geosci. Remote Sens. Lett., 8, 19-23, https://doi.org/10.1109/LGRS.2010.2049982, 2011.

---

## Author Comment (AC2) · 7 Dec 2018

The manuscript describes a methodology to discriminate between aerosol and cloud layers from CALIOP/CALIPSO lidar Level 2 data based on the high dimensional Fuzzy K-Means Cluster Analysis. The argument for sure is a good fit for the journal but some parts are not clear, probably suffering from hasty writing and need improvements before final publication. Moreover, other tests should be performed to improve scientific significance and clarity. I am however confident that the authors will brilliantly address all the issues I raised.

Thanks for reviewing the paper and giving valuable feedback. It is very hard to validate the operational algorithm at global scale, because we know of no existing global in-situ data set that could be used for the task. A comparison between different classification schemes used by active and passive sensor has been done in previous work (Stubenrauch et al., 2013). However, as active sensors profile the full vertical extent of the atmosphere, it remains quite difficult to compare classification results with passive sensors that, at best, only measure the properties of a single layer. (More often, properties of multiple layers are convolved into a single set of measurements, and thus tasks such as separately classifying cirrus clouds and boundary layer aerosols within the same pixel are extremely challenging retrievals for passive sensors.) Furthermore, comparisons between different algorithms have not yet been performed. Similar to the comparison between passive and active sensors, it's hard to determine how accurate the algorithms are (see our previous comments about the use of synthetic data), but by combining data from multiple sensors we can estimate upper and lower boundaries for cloud and aerosol distributions over the globe, and these values give a distribution range to guide modelers. Similarly, the comparison between supervised and unsupervised algorithms can also give upper and lower boundaries for precision to guide modelers, instrument developers, and data processors. To address the referee's concerns, we add detailed statements in the introduction to clarify these points. As we say in the conclusions of the original draft, the purpose of this study is "to validate the performance of the cloud-aerosol discrimination (CAD) algorithm used in the standard processing", and we are not suggesting FKM as a replacement for COCA. To this end we also added more introduction about the importance of discriminating between clouds and aerosols, and described the benefits for the study for different user communities.

Major Comments:

The FKM clustering methodology is well described and totally makes sense. But, as stated in the introduction, the FKM method is used to validate the result of V4 CAD algorithm and to better understand the classification, identifying the crucial parameters. It looks like that all the produced efforts have a very low return on investment. The V4 CAD is not validated vs. a reference dataset, i.e. using a synthetic lidar data where all the aerosol and cloud properties are well known and controlled, but with respect to another methodology that have comparable uncertainties.

While the use of synthetic data as an evaluation tool is generally an excellent and highly effective strategy, in the case of discriminating clouds from aerosols it's also especially hard to implement in a useful way. For ~90% of the cases, cloud and aerosol properties are very well separated and reliable classifications can be made using a single wavelength elastic backscatter lidar (e.g., CATS; see https://cats.gsfc.nasa.gov/media/docs/CATS_QS_L2O_Layer_3.00.pdf). In these cases, unambiguous synthetic data can be used to weed out those algorithms that are obviously deficient. But the remaining cases fall into the cloud-aerosol overlap region (see Liu et al., 2009), and for these layers even the most sophisticated human observers cannot always agree on the correct partitioning (e.g., see Koren et al., 2007; Tackett and Di Girolamo, 2009; Varnai and Marshak, 2011; Balmes and Fu, 2018). These especially difficult cases include separating thin cirrus from lofted Asian dusts, separating evaporating water cloud filaments from the surrounding aerosols in the marine boundary layer, and separating fresh volcanic ash from cirrus. Given the measurements available on the CALIPSO platform, the classification of these targets is always subject to some uncertainty. So, yes, we certainly could create "synthetic lidar data where all the aerosol and cloud properties are well known and controlled" and compare the classifications obtained from the CALIPSO operational CAD algorithm (COCA) and the FKM algorithm. And by using this synthetic data to compare algorithm outputs versus "truth" we could perhaps choose an algorithm that best confirms our own prejudices; but whether that algorithm was actually delivering the correct classifications in the really hard cases would still be an open question.

Moreover, It is completely missing an analysis on who is really using those data, i.e. climatologists, modelers..., and why it is critical to discriminate (defining a level of precision) between aerosols and clouds (and their subtypes). For example, how much is it the actual precision of the current operational V4 CAD algorithm in classifying the aerosol and cloud layers ? The final users are ok with this accuracy? Which benefits will be obtained reducing the misclassification? How the FKM will be used or implemented to reduce the V4 CAD misclassification?

While this would certainly be interesting information, this kind of detailed analysis lies well beyond the scope of this paper. (Simply counting up the number of different CALIPSO data user communities that make use of the CAD scores we provide would likely lead to some fascinating (and perhaps surprising!) insights.) In this paper, our goal is limited to providing a performance assessment of the current CALIPSO operational CAD algorithm.

In the manuscript is only marginally discussed why January 2008 measurement are a representative data sample. How the results are impacted changing the analyzed dataset?

As one-month data is enough for the purpose of the study, we just randomly choose one month. For different dataset, the class number and the fuzzy exponent may be different, but classification results on cloud and aerosol should not be too different in theory. In reality, for different season, different features occur which may slightly impact the sample of classes and thus the results. The paper just focuses on the first step of comparison and didn't go further. We mentioned this in the data preparation and added a summary of future work at the end of conclusion.

The number of classes is predefined (2 or 3) after analyzing Figure 3. However, in operational contexts, some data subsets might belong only to two classes. FKM still will fill with observation the class that should be empty. Is there a reason why the authors used the FKM cluster analysis instead of some self-selecting class methods, i.e. MeanShift clustering (Cheng, Yizong. "Mean shift, mode seeking, and clustering." IEEE transactions on pattern analysis and machine intelligence 17.8 (1995): 790-799) or classification algorithms as AdaBoost (Hu, Weiming, Wei Hu, and Steve Maybank.

"AdaBoost-based algorithm for network intrusion detection." IEEE Transactions on Sys- tems, Man, and Cybernetics, Part B (Cybernetics) 38.2 (2008): 577-583) ?

I think both Meanshift and Adaboost are very good algorithms to do clustering too. There are so many clustering methods (more than 100 maybe), supervised or unsupervised, connectivity-based, centroid based, density based and distribution clustering, we only try the Fuzzy K-means out, which is one of unsupervised centroid method that produces a membership which (between 0 and 1) is represent the probability of belonging to one class and is comparable to the official CAD scores (between -100 to 100, probability belong to one and the other). And also the shape of multi-dimensional observations of cloud and aerosol are suitable for centroid based algorithms. Last, the FKM unsupervised approach is quite different from the highly supervised method used to train the operational algorithm, is what we need for the comparison and the objective of the study.

Density based algorithms such as Meanshift expect some kind of density drop to detect cluster borders. Mean-shift is usually slower than k-Means. Besides that, the applicability of the mean-shift algorithm to multidimensional data is hindered by the unsmooth behavior of the kernel density estimate, which results in over-fragmentation of cluster tails (Achert et al. 2006). Clouds have two centers (ice and water) and aerosols may also have several sub-centers (e.g., dust and biomass burning), so a density based algorithm may not suitable for this classification in my opinion. Also, according to Kaur and Chawla (2015), FCM has higher accuracy compared to the Meanshift. AdaBoost is a machine learning method, and more complicated to understand. While using it may resolve the problem for FKM weighting problems in some future study, at the moment we want an easier understand method that is distinctly different from the COCA method investigate different algorithm inputs on the classifications. In the future we will consider to doing some machine learning classifications, but may not choose AdaBoost.

The random initialization of the centroids is a well-known problem as the initial centroid selection not only influences the efficiency of the algorithm, but also the number of relative iterations (and consequently the needed time machine). Some optimal centroid selection techniques can be found in Nazeer, K.A. Sebastian, M.P Clustering biological data using enhanced k-means algorithm". In: Electronic Engineering and Computing Technology, Springer, 2010, pp. 433–442 (chapter 37)

The flowchart is wrong in previous version of manuscript. We have a loop to choose the best initiation and outcome results in FKM algorithm. With the loop to choose the best initiation, the larger the number of loops, the better the resulting clusters will be, but this is not time efficient. In application to real data, we have not yet found that using a larger number of loops will consistently improve the classification accuracies for the CALIPSO level 2 observations.

Many thanks to the reviewer for introducing us to an efficient way to save the relative iteration number and time.

Specific comments:

Line 27 Pag. 1 Please add also "geometrical properties"

We added it.

Line 15 Pag. 5 How the random initialization influence the final result? I don't recall any section where this issue is discussed. Are the results consistent with the random initialization?

We misrepresented our algorithm, and so we modified our flowchart in Figure 1. As we do a loop to choose the best random initialization, outcome results do not change due to initiation as long as the iteration number and the loop number for selecting initiation are big enough.

Line 16 Pag. 5 the authors mean Equations 2, 3 and 4?

We corrected them.

Figure 1: Third step it should be Eq. 6 and 7

We corrected them.

Line 2 Pag. 7: I am not sure that latitude is not useful to discriminate, as clouds at 16 km at polar latitudes may rise a flag, as cirrus clouds below 9km in the equatorial and tropical regions

The region (i.e. latitude) and season information are of course useful auxiliary information because they can indicate the sources of particles and the dynamics of the atmosphere. The others are directly measured optical information of the particles due to their scattering nature. In the future, we can train and apply the FKM method at local scales, which could be a way to improve the current classifications.

Figure 3: labels are difficult to read. The picture in the middle shows "NCE" that is not previously defined.

We selected the bold font to the labels so as to see the label easier and changed the "NCE" to "MPE".

Line 14 Pag 11: please rephrase "water clouds. For these water clouds".

We rephrased it.

Figure 4: it is very hard to see the zone of interest (smoke and cloud). Maybe reduce the vertical scale from 0 to 20 km?

We modified it to 20km.

Line 15 Pag 17 please read "We saw" instead of "We see"

We corrected it.

Paragraphs 3.4 a, 3.4 b and 3.4 c. How the authors assume that the layer are pure dust, smoke and ash respectively? Is there any other ancillary measurement that shows without any doubt the aerosol layer composition?

This comment highlights one of the major difficulties in validating a global data set acquired by a first-of-its-kind active sensor: coincident measurements of interesting phenomena are extremely difficult to come by! For these events, we tracked these plumes by eye according to the event's location, time period and our experience in evaluating spatial distributions and layer optical features (depolarization, color ratio and backscatter). This is very accurate though.

Section 4. Figure 13 is not very intuitive and it is difficult to get meaningful information from it . It might be interesting to replace it (or add) the Screen Plot and the loading factors as barplot as showed in https://doi.org/10.1175/JTECH-D-15-0085.1.

The figure includes a lot of information compared to the barplot, but we did not explain it well. We have now added more explanation about the figures and added the color bar.

Line 4 Pag. 34: Even if the FKM Cluster Analysis closely replicate the CAD V4 operational algorithm, it is not validate it (see main comment section)

We changed the "validation" to "comparison". We explained more in the paper that the comparison between algorithms can set up boundaries for the uncertainness due to different algorithms

Line 18 Pag. 35. FKM it is a time consuming algorithm because setting up random centroids can slow down the convergence process and in some cases can produce as result sub-optimal centroids virtual centroids (i.e. not corresponding to any observational measurement). See Main Comments section.

Yes, we added more details to the related domain to clarify the reason for FKM "time consuming". We modify the algorithm description in the paper and in Figure 1.

**References**

Achtert, E.; Böhm, C.; Kröger, P. (2006). "DeLi-Clu: Boosting Robustness, Completeness, Usability, and Efficiency of Hierarchical Clustering by a Closest Pair Ranking". LNCS: Advances in Knowledge Discovery and Data Mining. Lecture Notes in Computer Science. 3918: 119–128. doi:10.1007/11731139_16. ISBN 978-3-540-33206-0.

Balmes, K. A., and Fu, Q.: An Investigation of Optically Very Thin Ice Clouds from Ground-Based ARM Raman Lidars, Atmosphere, 9, 445; doi:10.3390/atmos9110445, 2018.

Brakhasi, F., Matkan, A., Hajeb, M., and Khoshelham, K.: Atmospheric scene classification using CALIPSO spaceborne lidar measurements in the Middle East and North Africa (MENA), and India, International Journal of Applied Earth Observation and Geoinformation, 73, 721-735, https://doi.org/10.1016/j.jag.2018.07.017, 2018.

Kaur, S. and Chawla S. (2015): Evaluation of Performance of Fuzzy C Means and Mean Shift based Segmentation for Multi-Spectral Images, International Journal of Computer Applications 120, 25-28.

Koren, I., Remer, L. A., Kaufman, Y. J., Rudich, Y., and Martins, J. V.: On the twilight zone between clouds and aerosols, Geophys. Res. Lett., 34, L08805, doi:10.1029/2007GL029253, 2007.

Liu, Z., Vaughan, M., Winker, D., Kittaka, C., Getzewich, B., Kuehn, R., Omar, A., Powell, K., Trepte, C., and Hostetler, C.: The CALIPSO lidar cloud and aerosol discrimination: Version 2 algorithm and initial assessment of performance, J. Atmos. Oceanic Technol., 26, 1198-1213, https://doi.org/10.1175/2009JTECHA1229.1, 2009.

Liu, Z., Kar, J., Zeng, S., Tackett, J., Vaughan, M., Avery, M., Pelon, J., Getzewich, B., Lee, K.-P., Magill, B., Omar, A., Lucker, P., Trepte, C., and Winker, D.: Discriminating Between Clouds and Aerosols in

the CALIOP Version 4.1 Data Products, Atmos. Meas. Tech. Discuss., https://doi.org/10.5194/amt-2018-190, in review, 2018.

Tackett, J. L., and Di Girolamo, L.: Enhanced aerosol backscatter adjacent to tropical trade wind clouds revealed by satellite-based lidar, Geophys. Res. Lett., 36, L14804, https://doi.org/10.1029/2009GL039264, 2009.

Varnai, T. and Marshak, A.: Global CALIPSO Observations of Aerosol Changes Near Clouds, IEEE Geosci. Remote Sens. Lett., 8, 19-23, https://doi.org/10.1109/LGRS.2010.2049982, 2011.

---

## Author Response (AR2)

[revised manuscript text omitted]

We thank the referees for their careful reading of our manuscript and their thoughtful and constructive comments. We have reproduced the referee's comments below (in black Times New Roman) and included our responses in-line (in blue Tahoma).

**Report #1**

I am happy that the authors addressed all my previous issues. There are some minor changes before publications that are listed below:

Line 24 Pag 3: Please read "Tesche" instead of "Teche"

Done

Pag 11 and Pag 12: Equations are not numbered.

Done

**Report #2**

There were significant changes to the manuscript and I thank the authors for putting in the time to make this a strong candidate for publication. There are only a few areas that could use a bit more clarification, and thus, I believe this study should be accepted with minor revisions. Such areas are mentioned below.

In page 5, 2nd paragraph where the COCA PDF development is explained: You mention the difference between the initial COCA PDF (4 dimensional) to the V3 COCA PDF (5 dimensional). A sentence about the differences in the V3 to V4 COCA PDFs is needed here since the results from CAD V3 to CAD V4 are compared later.

We added the following sentences:

"However, while the V3 and V4 algorithms both use five independent measurements, the numerical values in the underlying PDFs are significantly different. The V3 PDFS were rendered obsolete by extensive changes to the V4 calibration algorithms (Kar et al., 2017; Getzewich et al., 2018; Vaughan et al., 2019) that required revising a number of the $<\beta'_{532}>$ probabilities and a global recalculation of the color ratio probabilities. As a result, the V4 CAD algorithm can now be applied in the stratosphere and to layers detected at single shot resolution and has greatly improved performance when identifying dense dust over the Taklimakan desert, lofted dust over Siberian and American Arctic regions, and high-altitude aerosols in the upper troposphere and lower stratosphere (Liu et al., 2019)."

In Fig 4 there are two patches of elevated backscatter, one between 36.86 and 71.55 around 15 km and the other south of -69.6 around 12-15 km. Both are classified as aerosol using CAD V4, but cloud using the 3 class FKM. Maybe include a sentence on the discrepancy.

This also brings me to my next comment. The feature south of -69.6 is indicated as aerosol in the 2 class FKM, but cloud in the 3 class FKM. Presumably, going from a 2-class to 3-class is analogous to going from aerosol/cloud scheme to aerosol/water cloud/ice cloud scheme, and would likely be so in binary k-means clustering. In that case, a pixel identified as cloud, would fall within ice cloud or water cloud when going to a three class scheme. This figure shows that assumption is

(1)

untrue, at least when using fuzzy logic. I think this phenomenon should be highlighted in the text, specifically as an example of the "twilight zone" between aerosol and diffuse high cloud.

We have added the following discussion:

"The differences between the FKM and V4 COCA classifications are most prominent for layers detected in the stratosphere (i.e., layers rendered in black in Fig. 4b). In the northern hemisphere, between 71.55° N and 36.86° N, a diffuse, weakly scattering layer is intermittently detected at altitudes between 12 km and 18 km. This layer most likely originated with the eruption of the Kasatochi volcano on 7 August 2008 (Krotkov et al., 2010). But while V4 COCA classifies these layers as aerosols, both FKM methods identify them as clouds. In the southern hemisphere south of 69.6° S, a faint polar stratospheric layer is detected continuously, with a mean base altitude at ~11 km and a mean top at ~15 km. Once again, the V4 COCA classifies this feature as a moderate-to-high confidence aerosol layer and the 3-class FKM classifies it as a high confidence cloud. However, unlike the northern hemisphere case, the 2-class FKM identifies this as low confidence aerosol. Correctly classifying stratospheric features that occupy the "twilight zone" between aerosols and highly tenuous clouds (Koren et al., 2007) is likely to be difficult for unsupervised learning methods, due to the extensive overlap in the available lidar observables for the two classes. Class separation is typically (though not always) more distinct within the troposphere."

Reference

Krotkov, N. A., Schoeberl, M. R., Morris, G. A., Carn, S., and Yang, K.: Dispersion and lifetime of the $SO_2$ cloud from the August 2008 Kasatochi eruption, J. Geophys. Res., 115, D00L20, https://doi.org/10.1029/2010JD013984, 2010.

Fig 7: for (c) and (f), I would choose to clip the scale at +/- 50%. Specifically, anything below 25% difference would not show on the plot as is, and, such a difference is not negligible. In fact, the authors have clipped the scale to +/- 10% when comparing zonal averages.

Done

For the discussion on table 2, maybe highlight that the biggest difference seems to be FKM indicating features as aerosol that were determined as cloud using COCA. To me, this suggests the "twilight zone" again. Since the authors are suggesting that a global FKM CAD algorithm is likely less accurate when compared to current CAD, a bit more discussion on the impacts of these features should be included somewhere.

We inserted the following additional discussion of the results shown in Table 2:

"While the discrepancies between the two techniques are pleasingly small, their root causes are still of some interest. For example, we note that the FKM algorithm shows a slight bias toward aerosols relative to the V4 COCA (a 2.4% bias for the 3-class FKM versus a 1.5% bias for the 2-class FKM). At present, we speculate that the bulk of these differences can be traced to the dichotomy between supervised (COCA) and unsupervised (FKM) learning techniques. Given scope and quality of the data currently available for use by the COCA and FKM methods, the correct classification of layers occupying the twilight zone separating clouds and aerosols remains somewhat uncertain, and hence different learning strategies are likely to come to different conclusions, even when provided the same evidence."

Figure 12 (b) is very difficult to read due to the very similar color choices. since 7 variables are being plotted, I'm not sure how much can be done, but it is nonetheless hard to analyze, especially when printed.

We have inserted a revised version of the figure that (we hope!) addresses these concerns.